# PROPER SCORING RULES FOR SURVIVAL ANALYSIS

## ABSTRACT

Survival analysis is the problem of estimating probability distributions for future events, which can be seen as a problem in uncertainty quantification. Although there are fundamental theories on strictly proper scoring rules for uncertainty quantification, little is known about those for survival analysis. In this paper, we investigate extensions of four major strictly proper scoring rules for survival analysis. Through the extensions, we discuss and clarify the assumptions arising from the discretization of the estimation of probability distributions. We also discuss the relationship between the existing algorithms and extended scoring rules, and we propose new algorithms based on our extensions of the scoring rules for survival analysis.

## 1 INTRODUCTION

The theory of *scoring rules* is a fundamental theory in statistical analysis, and it is widely used in uncertainty quantification (see, e.g., Mura et al. (2008); Parmigiani & Inoue (2009); Benedetti (2010); Schlag et al. (2015)). Suppose that there is a random variable $Y$ whose cumulative distribution function (CDF) is $F_Y$. Given an estimation $\hat{F}_Y$ of $F_Y$ and a single sample $y$ obtained from $Y$, a scoring rule $S(\hat{F}_Y, y)$ is a function that returns an evaluation score for $\hat{F}_Y$ based on $y$. Since $\hat{F}_Y$ is a CDF and $y$ is a single sample of $Y$, it is not straightforward to choose an appropriate scoring rule $S(\hat{F}_Y, y)$. The theory of scoring rules suggests *strictly proper* scoring rules that can be used to recover the true probability distribution $F_Y$ by optimizing the scoring rules. This theory shows that there are infinitely many strictly proper scoring rules, and examples of them include the pinball loss, the logarithmic score, the Brier score, and the ranked probability score (see, e.g., Gneiting & Raftery (2007) for the definitions of these scoring rules).

*Survival analysis*, which is also known as *time-to-event analysis*, can be seen a problem in uncertainty quantification. Despite the long history of research on survival analysis (see, e.g., Wang et al. (2019) for a comprehensive survey), little is known about the strictly proper scoring rules for survival analysis. Therefore, this paper investigates extensions of these scoring rules for survival analysis.

Survival analysis is the problem of estimating probability distributions for future events. In healthcare applications, an event usually corresponds to an undesirable event for a patient (e.g., a death or the onset of disease). The time between a well-defined starting point and the occurrence of an event is called the *survival time* or *event time*. Survival analysis has important applications in many fields such as credit scoring (Dirick et al., 2017) and fraud detection (Zheng et al., 2019) as well as healthcare. Although we discuss survival analysis in the context of healthcare applications, we can use the extended scoring rules for any other applications.

Datasets for survival analysis are *censored*, which means that events of interest might not be observed for a number of data points. This may be due to either the limited observation time window or missing traces caused by other irrelevant events. In this paper, we consider only *right censored* data, which is a widely studied problem setting in survival analysis. The exact event time of a right censored data point is unknown; we know only that the event had not happened up to a certain time for the data point. The time between a well-defined starting point and the last observation time of a right censored data point is called the *censoring time*.

One of the classical methods for survival analysis is the Kaplan-Meier estimator (Kaplan & Meier, 1958). It is a non-parametric method for estimating the probability distribution of survival times as a survival function $\kappa(t)$, where the value $\kappa(t)$ represents the *survival rate* at time $t$ (i.e., the ratio of

the patients who survived at time $t$). By definition, $\kappa(0) = 1$ and $\kappa(t)$ is a monotonically decreasing function.

Since there are many applications that require an estimate of the survival function for each patient rather than the overall survival function $\kappa(t)$ for all patients, many algorithms have been proposed. In particular, many neural network models have been proposed (e.g., (Lee et al., 2018; Avati et al., 2019; Ren et al., 2019; Kamran & Wiens, 2021; Tjandra et al., 2021)).

A problem with these neural network models is that most of them are not based on the theory of scoring rules except for (Rindt et al., 2022). Since we cannot directly use a known scoring rule due to censoring in survival analysis, the state-of-the-art neural network models for survival analysis use their own custom loss functions instead. Even though these custom loss functions can be seen as variants of known scoring rules, they are not proven to be strictly proper for survival analysis in terms of the theory of scoring rules.

We review variants of scoring rules used in survival analysis with respect to the four major strictly proper scoring rules.

- **Pinball loss.** Portnoy's estimator (Portnoy, 2003), which is a variant of the pinball loss, has been used in quantile regression-based survival analysis (Portnoy, 2003; Neocleous et al., 2006; Pearce et al., 2022). However, it is unknown if Portnoy's estimator is proper or not.

- **Logarithmic score.** Rindt et al. (2022) proved that a variant of the logarithmic score is strictly proper for survival analysis. This variant has been used in the loss function of many neural network models (e.g., (Lee et al., 2018; Avati et al., 2019; Ren et al., 2019; Kamran & Wiens, 2021; Kvamme & Borgan, 2021; Tjandra et al., 2021)). However, most of them use this variant in *part* of the loss functions, and these loss functions are used without the proof of properness.

- **Brier score.** The IPCW Brier score (Graf et al., 1999) and integrated Brier score (Graf et al., 1999) are widely used in survival analysis (e.g., (Kvamme et al., 2019; Haider et al., 2020; Han et al., 2021; Zhong et al., 2021)) as variants of the Brier score. However, Rindt et al. (2022) show that neither of them are not proper in terms of the theory of scoring rules.

- **Ranked probability score.** Variants of the ranked probability score have been proposed in (Avati et al., 2019; Kamran & Wiens, 2021), but (Rindt et al., 2022) show that they are not proper in terms of the theory of scoring rules.

**Our contributions.** We analyze survival analysis through the lens of the theory of scoring rules. First, we prove that Portnoy's estimator, which is an extension of the pinball loss, is proper under certain conditions. This result underpins the grid-search algorithm (Portnoy, 2003; Neocleous et al., 2006) and the CQRNN algorithm (Pearce et al., 2022), which is based on the expectation maximization (EM) algorithm. Second, we show another proof for an extension of the logarithmic score. This scoring rule has already been proven to be strictly proper in (Rindt et al., 2022), but our proof clarifies the implicit assumption in the proof. Third, we show that there are two other proper scoring rules for survival analysis under certain conditions by extending the Brier score and the ranked probability score. By using these extended scoring rules, we construct two new algorithms by using the EM algorithm.

## 2 RELATED WORK

Survival analysis has been traditionally studied under the *proportional hazard assumption*. Its seminal work is the Cox model (Cox, 1972), and many other prediction models have been proposed under this strong assumption. See, e.g., Wang et al. (2019) for a comprehensive survey of the prediction models based on this assumption. Since we do not require the theory of scoring rules under this assumption, we consider survival analysis *without* this assumption. Note that most of the state-of-the-art neural network models for survival analysis do not use this assumption.

Regarding evaluation metrics for survival analysis, the concordance index (C-index) (Harrell et al., 1982) has been widely used under the proportional hazard assumption. Some variants of the C-index (Antolini et al., 2005; Uno et al., 2011) are proposed for survival analysis without the proportional hazard assumption. However, they are proven to not be proper in terms of the theory of

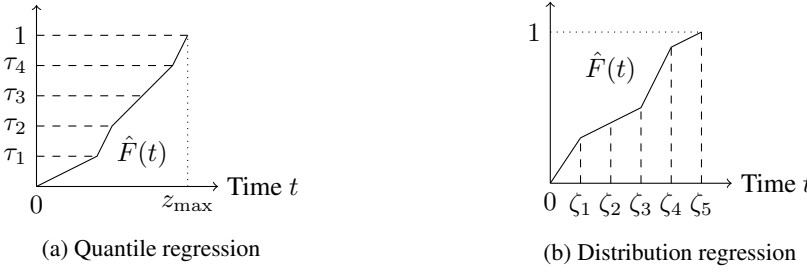

(a) Quantile regression

(b) Distribution regression

Figure 1: Two types of discretization of probability distribution $\hat{F}(t)$ with $B = 5$

scoring rules (Blanche et al., 2018; Rindt et al., 2022). Therefore, we do not use these variants of the C-index in this paper. We also note that Sonabend et al. (2022) discuss the problems of using these variants of the C-index in survival analysis.

## 3 PRELIMINARIES

We define notation here before showing the extensions of the scoring rules for survival analysis. Unless otherwise stated, we consider a single patient $x$, and let $T$ and $C$ be random variables for the event time and censoring time of this patient, respectively. Let $t \sim T$ and $c \sim C$ be samples obtained from $T$ and $C$, respectively. We assume that $t$ and $c$ are positive real values (i.e., $t \in \mathbb{R}^+$ and $c \in \mathbb{R}^+$). In survival analysis, we can observe only the minimum $z = \min\{t, c\}$, and we use $\delta = \mathbb{1}(t \leq c)$ to indicate whether $z$ represents the true event time (i.e., $\delta = 1$ means $z$ is uncensored, and $z = t$) or $z$ represents the censoring time (i.e., $\delta = 0$ means $z$ is censored, and $z = c$). In this paper, a pair of samples $(t, c)$ is often represented as a pair of values $(z, \delta)$ to emphasize that we can observe only one of $t$ and $c$. We assume that there exists $z_{\max} > 0$ such that $0 < z \leq z_{\max}$, which means that we have prior knowledge that $z$ is at most $z_{\max}$. Let $F(t)$ be the CDF of $T$, which is defined as $F(t) = \Pr(T \leq t)$. By the definition of $F(t)$, we have $F(0) = 0$, and we can represent the probability that the true event time is between $t_1$ and $t_2$ by $\Pr(t_1 < T \leq t_2) = F(t_2) - F(t_1)$.

Survival analysis is the problem of estimating the $\hat{F}(t)$ of the true CDF $F(t)$. For simplicity, we assume that both $F(t)$ and $\hat{F}(t)$ are monotonically increasing continuous functions. This means that $F(t_1) < F(t_2)$ holds if and only if $0 \leq t_1 < t_2 < \infty$. This assumption enables us to calculate $F(t)$ for any time $0 \leq t < \infty$ and to calculate $F^{-1}(\tau)$ for any quantile level $0 \leq \tau \leq 1$. When we estimate $\hat{F}(t)$ by using a neural network, we usually discretize $p = \hat{F}(t)$ along with the $p$-axis or the $t$-axis as shown in Fig. 1. In quantile regression-based survival analysis, $p = \hat{F}(t)$ is discretized along the $p$-axis, $\hat{F}^{-1}(\tau_i)$ is estimated for $0 = \tau_0 < \tau_1 < \cdots < \tau_{B-1} < \tau_B = 1$, and we assume that $\hat{F}^{-1}(\tau_0) = 0$ and $\hat{F}^{-1}(\tau_B) = z_{\max}$. In distribution regression-based survival analysis, $p = \hat{F}(t)$ is discretized along the $t$-axis, $\hat{F}(\zeta_i)$ is estimated for $0 = \zeta_0 < \zeta_1 < \cdots < \zeta_{B-1} < \zeta_B = z_{\max}$, and we assume that $\hat{F}(\zeta_0) = 0$ and $\hat{F}(\zeta_B) = 1$.

Throughout this paper we assume that the censoring time and the event time are independent of each other given a feature vector of patient $x$. This assumption is widely used in survival analysis, and this assumption is represented as

**Assumption 3.1.** $T \perp\!\!\!\perp C | X$.

Note that he Kaplan-Meier estimator (Kaplan & Meier, 1958), which is a classical non-parametric method for survival analysis, uses this assumption. D-calibration (Haider et al., 2020), which is one of the widely used metrics in survival analysis, also uses this assumption. We can find examples of the other stronger assumptions (e.g., unconditionally random right censoring) used in survival analysis in (Peng, 2021).

# 4 PROPER SCORING RULES FOR SURVIVAL ANALYSIS

We briefly review the theory of scoring rules for uncertainty quantification. Let $Y$ be a random variable, and let $F_Y(y)$ be its CDF, which is defined as $F_Y(y) = \Pr(Y \leq y)$. A *scoring rule* is a function $S(\hat{F}_Y, y)$ that returns a real value for inputs $\hat{F}_Y$ and $y$, where $\hat{F}_Y$ is an estimation of $F_Y$ and $y$ is a sample obtained from $Y$. In this paper, we consider *negatively-oriented* scoring rules. Therefore, the inequality $S(\hat{F}_1, y) < S(\hat{F}_2, y)$ means that $\hat{F}_1$ is a better estimation than $\hat{F}_2$. We can interpret the scoring rule $S(\hat{F}_Y, y)$ as a penalty function for the misestimation of $\hat{F}_Y$ for a sample $y$.

The *proper* and *strictly proper* scoring rules are defined by using the expected score of a scoring rule, which can be written as

$$\tilde{S}(\hat{F}_Y; Y) = \mathbb{E}_{y \sim Y}[S(\hat{F}_Y, y)].$$

**Definition 4.1.** *A scoring rule $S(\hat{F}_Y, y)$ is* proper *if $\tilde{S}(\hat{F}_Y; Y) \geq \tilde{S}(F_Y; Y)$ holds.*

**Definition 4.2.** *A scoring rule $S(\hat{F}_Y, y)$ is* strictly proper *if $\tilde{S}(\hat{F}_Y; Y) \geq \tilde{S}(F_Y; Y)$ holds and the equality holds only when $\hat{F}_Y = F_Y$.*

These definitions ask a proper scoring rule to satisfy that the score $S(\hat{F}_Y, y)$ for estimation $\hat{F}_Y$ is always at least $S(F_Y, y)$ for true CDF $F_Y$, and the score is minimized if $\hat{F}_Y = F_Y$. This is a natural property that any scoring rule should satisfy, and this means that we can recover the true $F_Y$ if we can minimize the score of a strictly proper scoring rule. The theory of scoring rules shows that there are infinitely many strictly proper scoring rules (see, e.g., Gneiting & Raftery (2007)).

Now we extend these definitions of the proper and strictly proper scoring rules for survival analysis. In survival analysis, the inputs of a scoring rule $S(\hat{F}, (z, \delta))$ are changed from $F_Y$ and $y$ to $F$ and $(z, \delta)$. The *proper* and *strictly proper* scoring rules are defined by using

$$\tilde{S}(\hat{F}; T, C) = \mathbb{E}_{(t,c) \sim (T,C)}[S(\hat{F}, (z, \delta))].$$

**Definition 4.3.** *A scoring rule $S(\hat{F}, (z, \delta))$ is* proper *if $\tilde{S}(\hat{F}; T, C) \geq \tilde{S}(F; T, C)$ holds.*

**Definition 4.4.** *A scoring rule $S(\hat{F}, (z, \delta))$ is* strictly proper *if $\tilde{S}(\hat{F}; T, C) \geq \tilde{S}(F; T, C)$ holds and the equality holds only when $\hat{F} = F$.*

Now we investigate the extensions of the scoring rules for survival analysis. In Sec. 4.1, we consider quantile regression and survival analysis based on quantile regression. In Secs. 4.2–4.4, we consider distribution regression and survival analysis based on distribution regression.

## 4.1 EXTENSION OF PINBALL LOSS

We first review quantile regression (Koenker & Bassett, 1978; Koenker & Hallock, 2001). Let $Y$ be a real-valued random variable and $F_Y$ be its CDF. In quantile regression, we estimate the $\tau$-th quantile of $Y$, which can be written as

$$F_Y^{-1}(\tau) = \inf\{y \mid F_Y(y) \geq \tau\}.$$

The *pinball loss* (Koenker & Bassett, 1978), which is also known as the *check function*, is a widely used scoring rule. The pinball loss for an estimation $\hat{F}_Y$ of $F_Y$ and a quantile level $\tau$ is defined as

$$S_{\text{Pinball}}(\hat{F}_Y, y; \tau) = \rho_\tau(\hat{F}_Y^{-1}(\tau), y) = \begin{cases} (1 - \tau)(\hat{F}_Y^{-1}(\tau) - y) & \text{if } \hat{F}_Y^{-1}(\tau) \geq y, \\ \tau(y - \hat{F}_Y^{-1}(\tau)) & \text{if } \hat{F}_Y^{-1}(\tau) < y. \end{cases} \quad (1)$$

Note that the pinball loss with $\tau = 0.5$ is equivalent to the mean absolute error (MAE) and it can be used to estimate the *median* (i.e., 0.5-th quantile) of $Y$. This means that the pinball loss is a generalization of MAE for any quantile level $\tau \in [0, 1]$. Note also that we include the quantile level $\tau$ in the notation $S_{\text{Pinball}}(\hat{F}_Y^{-1}, y; \tau)$ to clarify that this scoring rule receives $\tau$ as an input.

It is known that the pinball loss is strictly proper (see e.g., (Gneiting & Raftery, 2007)), which means that we have

$$\mathbb{E}_{y \sim Y}[S_{\text{Pinball}}(\hat{F}_Y, y; \tau)] \geq \mathbb{E}_{y \sim Y}[S_{\text{Pinball}}(F_Y, y; \tau)],$$

and the equality holds only when $\hat{F}_Y^{-1}(\tau) = F_Y^{-1}(\tau)$ by Definition 4.2. Therefore, quantile regression can be formulated as the problem of computing

$$\arg\min_{\hat{F}_Y} \mathbb{E}_{y \sim Y}[S_{\text{Pinball}}(\hat{F}_Y, y; \tau)].$$

As an extension of the pinball loss for quantile regression-based survival analysis, *Portnoy's estimator* is proposed in Portnoy (2003), which is defined as

$$S_{\text{Portnoy}}(\hat{F}, (z, \delta); w, \tau) = \begin{cases} \rho_\tau(\hat{F}^{-1}(\tau), z) & \text{if } \delta = 1, \\ w\rho_\tau(\hat{F}^{-1}(\tau), z) + (1-w)\rho_\tau(\hat{F}^{-1}(\tau), z_\infty) & \text{if } \delta = 0, \end{cases} \quad (2)$$

where $\rho_\tau$ is the pinball loss defined in Eq. (1), $w$ is a weight parameter to control the balance between two pinball loss terms, and $z_\infty$ is any constant such that $z_\infty > z_{\max}$. In Portnoy's estimator, we can set an arbitrary constant $0 \le w \le 1$ for the parameter $w$ if $\tau_c > \tau$, where $\tau_c = \Pr(t \le c) = F(c)$, but we have to set $w = \Pr(F(c) < F(t) \le \tau | t > c) = (\tau - \tau_c)/(1 - \tau_c)$ otherwise (i.e., $\tau_c \le \tau$). Since we do not know the true value $\tau_c = F(c)$, we have to resolve this problem to use this estimator.

Before showing how to resolve this problem, we prove that this estimator is proper under the condition that $w$ is correct. Note that this is the first result for the quantile regression-based survival analysis in terms of the theory of scoring rules.

**Theorem 4.5.** *Portnoy's estimator is proper under the condition that $w$ is correct.*

*Proof.* The proof is given in Appendix A.1. □

This theorem means that the crucial part of Portnoy's estimator is to set an appropriate value for $w$, and this theorem ensures that we can recover the true probability distribution $F^{-1}$ by minimizing Eq. (2) if $w$ is correct.

Now we discuss how to set parameter $w$ in Portnoy's estimator. First, we emphasize that we cannot avoid the dependence on $F(c)$ in the definition of any of the scoring rules for survival analysis due to the discretization of $\hat{F}$. Even if we know the true value $F^{-1}(\tau_i)$ for all $\{\tau_i\}_{i=0}^B$, we cannot compute $F(c)$ because $c$ is not always contained in $\{F^{-1}(\tau_i)\}_{i=0}^B$. The best we can do is to find quantile levels $\tau_i$ and $\tau_{i+1}$ such that $F^{-1}(\tau_i) < c \le F^{-1}(\tau_{i+1})$ by using the assumption that $F$ is a monotonically increasing function. Note that, even if we could find such $\tau_i$ and $\tau_{i+1}$, we would not be able to calculate some important probabilities such as $\Pr(c < t \le F^{-1}(\tau_{i+1})) = \tau_{i+1} - F(c)$. Therefore, we usually mitigate this problem by using a large $B$, which enables us to assume, for example, $F(\tau_{i+1}) - F(\tau_i) \approx 0$ for all $i$.

Even if we use a large $B$ to assume that we can find the quantile level $\tau_c'$ such that $c \approx F^{-1}(\tau_c')$ for any $c$, the problem that we do not know the true $F^{-1}$ remains. One of the approaches to tackling this problem is the grid search algorithm (Portnoy, 2003; Neocleous et al., 2006). In this algorithm, we use a sufficiently large $B$, and we estimate $\hat{F}^{-1}(\tau_i)$ of $F^{-1}(\tau_i)$ in the increasing order of $i = 0, 1, \ldots, B$. Suppose that we have estimated $\{\hat{F}^{-1}(\tau_i)\}_{i=0}^{j-1}$ and we are going to estimate $\hat{F}^{-1}(\tau_j)$. The key idea of this algorithm is that we can find $\tau_c' \in \{\tau_i\}_{i=0}^{j-1}$ such that $c \approx \hat{F}^{-1}(\tau_c')$ if $\tau_c = F(c) < \tau_j$. If we can find such $\tau_c'$, we estimate $w$ by using $\tau_c' \approx \tau_c$. If we cannot find such $\tau_c'$, this algorithm assumes that $\tau_c > \tau_j$ and we use an arbitrary constant $0 \le w \le 1$. Portnoy (2003) discuss that this algorithm is analogous to the Kaplan-Meier estimator, and their theoretical analysis (Portnoy, 2003; Neocleous et al., 2006) proves that Portnoy's estimator combined with linear regression can recover the true probability distribution $F$.

As for another approach, Pearce et al. (2022) propose the CQRNN algorithm, which combines a neural network and the EM algorithm. Unlike the grid search algorithm, this algorithm estimates $\{\hat{F}^{-1}(\tau_i)\}_{i=0}^B$ simultaneously by using a neural network. This algorithm starts with an arbitrary initial estimation $\hat{F}$, and the parameter $w$ is estimated by using $\hat{F}$. Then, this algorithm updates $\hat{F}$ by using the estimation $\hat{w}$ of $w$, and it repeats this alternative estimation of $\hat{F}$ and $\hat{w}$ until these values converge. This EM algorithm can be implemented for "free" according to (Pearce et al., 2022), which means that we can implement it easily in the computation of the loss function of a neural network training algorithm and we do not need to construct two separate neural network models for estimating $\hat{F}$ and $\hat{w}$. The experimental evaluation in (Pearce et al., 2022) shows that the CQRNN algorithm performs the best among the quantile regression-based survival analysis models.

## 4.2 EXTENSION OF LOGARITHMIC SCORE

While we estimate $\{\hat{F}^{-1}(\tau_i)\}_{i=0}^B$ in quantile regression, we consider distribution regression, in which we estimate $\{\hat{F}(\zeta_i)\}_{i=0}^B$. For distribution regression, the logarithmic score (Good, 1952) is known as one of the strictly proper scoring rules, and it is defined as

$$
\begin{aligned}
S_{\log}(\hat{F}, y; \{\zeta_i\}_{i=0}^B) &= -\sum_{i=0}^{B-1} \mathbb{1}(\zeta_i < y \le \zeta_{i+1}) \log(\hat{F}(\zeta_{i+1}) - \hat{F}(\zeta_i)) \\
&= -\sum_{i=0}^{B-1} \mathbb{1}(\zeta_i < y \le \zeta_{i+1}) \log \hat{f}_i,
\end{aligned}
\tag{3}
$$

where $\hat{f}_i = \hat{F}(\zeta_{i+1}) - \hat{F}(\zeta_i)$ for $i = 0, 1, \ldots, B-1$.

We extend this logarithmic score for distribution regression-based survival analysis as

$$
\begin{aligned}
&S_{\text{Cen}-\log}(\hat{F}, (z, \delta); w, \{\zeta_i\}_{i=0}^B) \\
&= -\sum_{i=0}^{B-1} \mathbb{1}(\zeta_i < z \le \zeta_{i+1}) \left( \delta \log \hat{f}_i + (1-\delta)(w \log \hat{f}_i + (1-w) \log(1 - \hat{F}(\zeta_{i+1}))) \right),
\end{aligned}
\tag{4}
$$

where $w = \Pr(c < t \le \zeta_{i+1} | t > c) = (F(\zeta_{i+1}) - F(c))/(1 - F(c))$. If $\delta = 1$, this scoring rule is equal to Eq. (3). Similar to Portnoy's estimator, we cannot set the parameter $w$ of this scoring rule because we do not know $F(\zeta_{i+1})$ and $F(c)$.

Even though we do not know the correct $w$, we prove that this scoring rule is strictly proper if the parameter $w$ is correct.

**Theorem 4.6.** *The scoring rule* $S_{\text{Cen}-\text{Log}}(\hat{F}, (z, \delta); w, \{\zeta_i\}_{i=0}^B)$ *is strictly proper if $w$ is correct.*

*Proof.* The proof is given in Appendix A.2. □

Similar to Portnoy's estimator, we can use both the grid-search algorithm and an EM algorithm similar to the CQRNN algorithm to estimate $w$. In addition, we show another simpler approach by using the observation that $w \approx 0$ if $B$ is large. If $B$ is large, $1 - F(c)$ is usually much larger than $F(\zeta_{i+1}) - F(c)$ (see Fig. 2(a)), and hence we have $w = (F(\zeta_{i+1}) - F(c))/(1 - F(c)) \approx 0$. Therefore, we obtain a simpler variant of $S_{\text{Cen}-\log}$ by setting $w = 0$:

$$
\begin{aligned}
&S_{\text{Cen}-\text{simple}-\log}(\hat{F}, (z, \delta); \{\zeta_i\}_{i=0}^B) \\
&= -\sum_{i=0}^{B-1} \mathbb{1}(\zeta_i < z \le \zeta_{i+1}) \left( \delta \log \hat{f}_i + (1-\delta) \log(1 - \hat{F}(\zeta_{i+1})) \right).
\end{aligned}
\tag{5}
$$

Furthermore, by increasing $B$ to infinity (i.e., $B \to \infty$), we obtain the continuous version of this scoring rule:

$$
S_{\text{Cen}-\text{cont}-\log}(\hat{F}, (z, \delta)) = -\delta \log \frac{d\hat{F}}{dt}(z) - (1-\delta) \log(1 - \hat{F}(z)),
$$

which is equal to the extension of the logarithmic score that is proven to be strictly proper in (Rindt et al., 2022). This clarifies that the proof in (Rindt et al., 2022) implicitly assumes that $B$ is sufficiently large.

## 4.3 EXTENSION OF BRIER SCORE

In distribution regression, the Brier score (Brier, 1950) is also known as a strictly proper scoring rule, which is defined as

$$
S_{\text{Brier}}(\hat{F}, y; \{\zeta_i\}_{i=0}^B) = \sum_{i=0}^{B-1} (\mathbb{1}(\zeta_i < y \le \zeta_{i+1}) - f_i)^2,
\tag{6}
$$

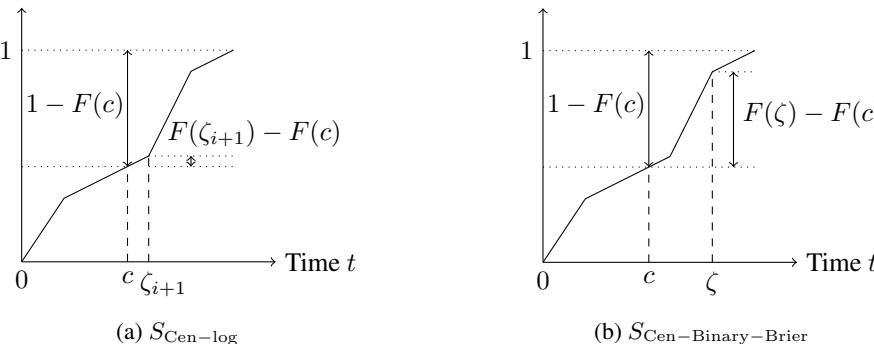

(a) $S_{\text{Cen}-\log}$           (b) $S_{\text{Cen}-\text{Binary}-\text{Brier}}$

Figure 2: Illustrations of computations of $w$

where $\hat{f}_i = \hat{F}(\zeta_{i+1}) - \hat{F}(\zeta_i)$ for $i = 0, 1, \ldots, B - 1$.

We extend this Brier score for distribution regression-based survival analysis as

$$S_{\text{Cen}-\text{Brier}}(\hat{F}, (z, \delta); \{w_i\}_{i=0}^{B-1}, \{\zeta_i\}_{i=0}^{B}) = \sum_{i=0}^{B-1} \left( w_i(1 - \hat{f}_i)^2 + (1 - w_i)\hat{f}_i^2 \right), \qquad (7)$$

where

$$w_i = \begin{cases} 0 & \text{if } \delta = 1 \text{ and } \zeta_{i+1} < z = t \\ 1 & \text{if } \delta = 1 \text{ and } \zeta_i < z = t \leq \zeta_{i+1} \\ 0 & \text{if } z \leq \zeta_i \\ (F(\zeta_{i+1}) - F(c))/(1 - F(c)) & \text{if } \delta = 0 \text{ and } \zeta_i < z = c \leq \zeta_{i+1} \\ f_j/(1 - F(c)) & \text{if } \delta = 0 \text{ and } \zeta_{i+1} < z = c. \end{cases}$$

If $\delta = 1$, it is easy to see that Eq. (7) is equal to Eq. (6).

We prove that this scoring rule is strictly proper if the set of parameters $\{w_i\}_{i=0}^{B-1}$ is correct.

**Theorem 4.7.** *The scoring rule $S_{\text{Cen}-\text{Brier}}(\hat{F}, (z, \delta); \{w_i\}_{i=0}^{B-1}, \{\zeta_i\}_{i=0}^{B})$ is strictly proper if $w_i$ is correct for all $i$.*

*Proof.* The proof is given in Appendix A.3. $\qquad \square$

We can use the EM algorithm similar to the CQRNN algorithm to estimate $w$. However, unlike Portnoy's estimator and the extension of the logarithmic score, we cannot use the grid-search algorithm in this extension of the Brier score because $w_i$ depends on $f_j$ such that $i < j$.

Note that each $w_i$ in this scoring rule is close to zero if $B$ is large and $\delta = 0$. However, since $w_i$s are designed to satisfy $\sum_i w_i = 1$, we cannot use the approximation $w_i \approx 0$ for this scoring rule.

## 4.4 EXTENSION OF RANKED PROBABILITY SCORE

The ranked probability score (RPS) is also known as a strictly proper scoring rule (see e.g., (Gneiting & Raftery, 2007)). It is defined as

$$S_{\text{RPS}}(\hat{F}, y) = \sum_{i=1}^{B} S_{\text{Binary}-\text{Brier}}(\hat{F}, y; \zeta_i), \qquad (8)$$

where $S_{\text{Binary}-\text{Brier}}$ is the binary version of $S_{\text{Brier}}$ (Eq. (6)) with single threshold $\zeta$:

$$S_{\text{Binary}-\text{Brier}}(\hat{F}, y; \zeta) = (\mathbb{1}(y \leq \zeta) - 1)^2. \qquad (9)$$

We extend this scoring rule for survival analysis:

$$S_{\text{Cen}-\text{RPS}}(\hat{F}, (z, \delta); \{w_i\}_{i=1}^{B-1}, \{\zeta_i\}_{i=1}^{B-1}) = \sum_{i=1}^{B-1} S_{\text{Cen}-\text{Binary}-\text{Brier}}(\hat{F}, (z, \delta); w_i, \zeta_i), \qquad (10)$$

where $S_{\text{Cen}-\text{Binary}-\text{Brier}}$ is the binary version of $S_{\text{Cen}-\text{Brier}}$ (Eq. (7)) with single threshold $\zeta$:

$$S_{\text{Cen}-\text{Binary}-\text{Brier}}(\hat{F}, (z, \delta); w, \zeta) = \begin{cases} \hat{F}(\zeta)^2 & \text{if } z > \zeta \\ (1 - \hat{F}(\zeta))^2 & \text{if } \delta = 1 \text{ and } z = t \leq \zeta \\ w(1 - \hat{F}(\zeta))^2 + (1 - w)\hat{F}(\zeta)^2 & \text{if } \delta = 0 \text{ and } z = c \leq \zeta, \end{cases}$$

where $w = (F(\zeta) - F(c))/(1 - F(c))$.

Since this scoring rule is just the sum of the binary version of Brier scores, it is straightforward to prove this theorem.

**Theorem 4.8.** *The scoring rule $S_{\text{Cen}-\text{RPS}}(\hat{F}, (z, \delta); \{w_i\}_{i=1}^{B-1}, \{\zeta_i\}_{i=1}^{B-1})$ is strictly proper if $w_i$ is correct for all $i$.*

Note that the scoring rule $S_{\text{Cen}-\text{Binary}-\text{Brier}}$ is analogous to Portnoy's estimator. The scoring rule $S_{\text{Cen}-\text{Binary}-\text{Brier}}$ is designed to estimate $\hat{F}(\zeta)$, where $\zeta$ is an input, and we use $F(c)$ and $\zeta$ to set $w$, whereas Portnoy's estimator is designed to estimate $\hat{F}^{-1}(\tau)$, where $\tau$ is an input, and we use $F(c)$ and $\tau$ to set $w$. As these two scoring rules are similar, we can use both the grid-search algorithm and an EM algorithm similar to the CQRNN algorithm for $S_{\text{Cen}-\text{RPS}}$.

Unlike $S_{\text{Cen}-\log}$ defined in Eq. (4), the parameter $w$ of the scoring rule $S_{\text{Cen}-\text{Binary}-\text{Brier}}$ is usually not close to zero, because $\zeta$ and $c$ are usually not close to each other as shown in Fig. 2(b). We note that the parameter $w$ of Portnoy's estimator is also not close to zero for a similar reason.

## 5 EVALUATION METRICS FOR SURVIVAL ANALYSIS

While we have discussed the extensions of the scoring rules as loss functions, we can use strictly proper scoring rules also for evaluation metrics. If we use a nonproper scoring rule $S_{\text{nonproper}}$ as an evaluation metric, a neural network model can find $\hat{F}$ such that

$$\mathbb{E}_{(t,c) \sim (T,C)}[S_{\text{nonproper}}(\hat{F}, (z, \delta))] < \mathbb{E}_{(t,c) \sim (T,C)}[S_{\text{nonproper}}(F, (z, \delta))]$$

by using $S_{\text{non}-\text{proper}}$ for the loss function. This suggests that we should avoid nonproper scoring rules as evaluation metrics, because we may obtain an over-optimized estimation $\hat{F}$, which has a lower score than $F$ in terms of the evaluation metric $S_{\text{nonproper}}$.

Among the extensions of the scoring rules for survival analysis, we can use only $S_{\text{Cen}-\text{simple}-\log}$ (Eq. (5)) as an evaluation metric for survival analysis, because the other scoring rules depend on the parameter $w$ or $\{w_i\}_{i=1}^{B-1}$. Note that this scoring rule $S_{\text{Cen}-\text{simple}-\log}$ is valid only if $B$ is sufficiently large. In Appendix B, we conducted experiments on choosing an appropriate $B$, and the results suggested using $B > 16$.

Regarding calibration metrics for survival analysis, while D-calibration (Haider et al., 2020) is widely used, we propose another metric for calibration, *KM-calibration*. We define this metric as

$$d_{\text{KM}-\text{cal}}(\kappa, \hat{F}_{\text{avg}}; \{\zeta_i\}_{i=0}^B) = d_{KL}(\kappa || 1 - \hat{F}_{\text{avg}}; \{\zeta_i\}_{i=0}^B) = \sum_{i=0}^{B-1} (p_i \log p_i - p_i \log q_i),$$

where $\kappa$ is the survival function estimated by using the Kaplan-Meier estimator (Kaplan & Meier, 1958), $\hat{F}_{\text{avg}}$ is the average of the estimated CDFs of all patients, $p_i = \kappa(\zeta_{i+1}) - \kappa(\zeta_i)$, and $q_i = (1 - \hat{F}_{\text{avg}}(\zeta_{i+1})) - (1 - \hat{F}_{\text{avg}}(\zeta_i))$. (In this computation, we assume that $\kappa(\zeta_B) = 0$.) This metric is the Kullback-Leibler divergence between $\kappa(t)$ and the average of the estimated survival function $1 - \hat{F}_{\text{avg}}(t)$. This metric is based on the observation that the model's predicted number of events within any time interval should be similar to the observed number (Goldstein et al., 2020).

We note here that calibration is particularly important for survival analysis especially in healthcare applications. If we use a prediction model that is miscalibrated, the predictions obtained from the miscalibrated model would be pessimistic or optimistic as a whole compared with the actual ones. If medical doctors were to make decisions on patient treatments on the basis of such a miscalibrated prediction model, the treatments could be harmful for patients because of the pessimistic or optimistic predictions. Calster et al. (2019) extensively discuss the importance of calibration in survival analysis.

Table 1: Prediction performances (lower is better) of extended scoring rules with $B = 32$

| Metric | Loss Function | flchain | prostateSurvival | support |
|---|---|---|---|---|
| $S_{\text{Cen}-\log-\text{simple}}$ | $S_{\text{Cen}-\log}$ | $1.5054 \pm 0.0508$ | $1.3608 \pm 0.0295$ | $1.8307 \pm 0.0452$ |
| | $S_{\text{Cen}-\text{Brier}}$ | $1.5137 \pm 0.0557$ | $1.3680 \pm 0.0291$ | $1.8467 \pm 0.0448$ |
| | $S_{\text{Cen}-\text{RPS}}$ | $1.6737 \pm 0.0821$ | $1.4821 \pm 0.0639$ | $2.1036 \pm 0.1012$ |
| | $S_{\text{Portnoy}}$ | $1.6641 \pm 0.0518$ | $1.4352 \pm 0.0420$ | $2.0645 \pm 0.0455$ |
| D-calibration | $S_{\text{Cen}-\log}$ | $0.0003 \pm 0.0001$ | $0.0001 \pm 0.0000$ | $0.0063 \pm 0.0009$ |
| | $S_{\text{Cen}-\text{Brier}}$ | $0.0004 \pm 0.0002$ | $0.0001 \pm 0.0000$ | $0.0071 \pm 0.0009$ |
| | $S_{\text{Cen}-\text{RPS}}$ | $0.0005 \pm 0.0003$ | $0.0010 \pm 0.0005$ | $0.0045 \pm 0.0011$ |
| | $S_{\text{Portnoy}}$ | $0.0071 \pm 0.0031$ | $0.0055 \pm 0.0041$ | $0.0237 \pm 0.0037$ |
| KM-calibration | $S_{\text{Cen}-\log}$ | $0.0206 \pm 0.0049$ | $0.0312 \pm 0.0084$ | $0.0299 \pm 0.0115$ |
| | $S_{\text{Cen}-\text{Brier}}$ | $0.0268 \pm 0.0071$ | $0.0324 \pm 0.0090$ | $0.0492 \pm 0.0125$ |
| | $S_{\text{Cen}-\text{RPS}}$ | $0.1553 \pm 0.0349$ | $0.5931 \pm 0.3846$ | $0.2668 \pm 0.1192$ |
| | $S_{\text{Portnoy}}$ | $0.0434 \pm 0.0067$ | $0.1895 \pm 0.1413$ | $0.0809 \pm 0.0381$ |

## 6 EXPERIMENTS

In our experiments, we used three datasets for the survival analysis from the packages in R R Core Team (2016): the flchain dataset Dispenzieri et al. (2012), which was obtained from the 'survival' package and contains 7874 data points (69.9% of which are censored), the prostateSurvival dataset (Lu-Yao et al., 2009), which was obtained from the 'asaur' package and contains 14294 data points (71.7% of which are censored), and the support dataset Knaus et al. (1995), which was obtained from the 'casebase' package and contains 9104 data points (31.9% of which are censored).

We compared the prediction performances of the extended scoring rules: $S_{\text{Cen}-\log}$ (Eq. (4)), $S_{\text{Cen}-\text{Brier}}$ (Eq. (7)), $S_{\text{Cen}-\text{RPS}}$ (Eq. (10)), and $S_{\text{Portnoy}}$ (Eq. (2)). We used a neural network model with $B = 32$ to estimate $\hat{F}$, and we combined it with the EM algorithm to estimate $w$ or $\{w_i\}_{i=0}^{B-1}$. This means that we used the CQRNN algorithm (Pearce et al., 2022), which is the state-of-the-art model for quantile regression-based survival analysis, for $S_{\text{Portnoy}}$. We used $S_{\text{Cen}-\log-\text{simple}}$ (Eq. (5)) as a metric for discrimination performce. We used D-calibration (Haider et al., 2020) and KM-calibration as metrics for calibration performance, where we used 20 bins of equal-length for D-calibration.

Table 1 shows the results, and each number shows the mean and standard deviation of the measurements of five-fold cross validation. These results showed that $S_{\text{Cen}-\log}$ and $S_{\text{Cen}-\text{Brier}}$ performed the best. Note that the former one is almost equal to the variant of the logarithmic score used in (Rindt et al., 2022), and that the latter one is our new extension of Brier score. Compared to these two scoring rules, the prediction performance of $S_{\text{Cen}-\text{RPS}}$ and $S_{\text{Portnoy}}$ were worse than expected and these results seemed to be not close to the true probability distribution, even though we prove that they are conditionally proper scoring rules. It is considered that the estimation $\hat{w}$ of parameter $w$ by the EM algorithm was not accurate enough to converge to the true probability distribution for $S_{\text{Cen}-\text{RPS}}$ and $S_{\text{Portnoy}}$. As we illustrate in Figure 2, the parameter $w$ of $S_{\text{Cen}-\text{RPS}}$ (and $S_{\text{Portnoy}}$) is usually not close to zero unlike $S_{\text{Cen}-\log}$, and this fact indicates that it was difficult to find correct $w$ for these two scoring rules.

## 7 CONCLUSION

We have discussed the extensions of the four scoring rules for survival analysis, and we have proved that these extensions are proper if the parameter $w$ or $\{w_i\}_{i=0}^{B-1}$ is correct. By using these scoring rules, we present neural network models combined with the EM algorithm to estimate the parameter, and our experiments showed that the models with $S_{\text{Cen}-\log}$ and $S_{\text{Cen}-\text{Brier}}$ performed the best. In addition, we clarified the hidden assumption in the proof of the variant of the logarithmic score for survival analysis (Rindt et al., 2022), which suggests us to use a sufficiently large $B$ when we use it as an evaluation metric. We believe that our approach of extending scoring rules for survival analysis can be used for many other known strictly proper scoring rules.

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

# A    PROOFS OF THEOREMS

We give proofs of the theorems, which are omitted from the main body of this paper.

## A.1    PORTNOY'S ESTIMATOR

We show a proof of Theorem 4.5.

*Proof.* We consider a fixed $c \sim C$, and we prove

$$\underset{t \sim T|C=c}{\mathbb{E}} [S_{\text{Portnoy}}(\hat{F}, (z, \delta); w, \tau)] \geq \underset{t \sim T|C=c}{\mathbb{E}} [S_{\text{Portnoy}}(F, (z, \delta); w, \tau)] \tag{11}$$

for these four cases separately.

- Case 1: $c \leq \min\{F^{-1}(\tau), \hat{F}^{-1}(\tau)\}$.

- Case 2: $\max\{F^{-1}(\tau), \hat{F}^{-1}(\tau)\} < c$.

- Case 3: $F^{-1}(\tau) < c \leq \hat{F}^{-1}(\tau)$.

- Case 4: $\hat{F}^{-1}(\tau) < c \leq F^{-1}(\tau)$.

Note that, if Inequality (11) holds for any $c \sim C$, we can marginalize the inequality with respect to $C$, and we can prove

$$\underset{t \sim T, c \sim C}{\mathbb{E}} [S_{\text{Portnoy}}(\hat{F}, (z, \delta); w, \tau)] \geq \underset{t \sim T, c \sim C}{\mathbb{E}} [S_{\text{Portnoy}}(F, (z, \delta); w, \tau)],$$

which means that $S_{\text{Portnoy}}(\hat{F}, (z, \delta); w, \tau)$ is proper. Therefore, we prove Inequality (11) for the four cases.

**Case 1.**    We prove the case for $c \leq \min\{F^{-1}(\tau), \hat{F}^{-1}(\tau)\}$. This means that $\tau_c \leq \tau$ and $w = (\tau - \tau_c)/(1 - \tau_c)$. Hence, we have

$$S_{\text{Portnoy}}(\hat{F}, (z, \delta); w, \tau) = \begin{cases} \rho_\tau(\hat{F}^{-1}(\tau), t) & \text{if } t \leq c, \\ w\rho_\tau(\hat{F}^{-1}(\tau), c) + (1 - w)\rho_\tau(\hat{F}^{-1}(\tau), z_\infty) & \text{if } t > c, \end{cases}$$

$$= \begin{cases} (1 - \tau)(\hat{F}^{-1}(\tau) - t) & \text{if } t \leq c, \\ -\tau_c(1 - \tau)(\hat{F}^{-1}(\tau) - t)/(1 - \tau_c) & \text{if } t > c. \end{cases}$$

By Assumption 3.1, we have $\Pr(t \leq c | C = c) = \Pr(t \leq c) = \tau_c$ and $\Pr(t > c | C = c) = 1 - \tau_c$. Hence, we have

$$\begin{aligned} \underset{t \sim T|C=c}{\mathbb{E}} [S_{\text{Portnoy}}(\hat{F}, (z, \delta); w, \tau)] &= \Pr(t \leq c | C = c)(1 - \tau)\hat{F}^{-1}(\tau) - (1 - \tau) \underset{t \sim T|C=c, t \leq c}{\mathbb{E}} [t] \\ &\quad - \Pr(t > c | C = c)\tau_c(1 - \tau)\hat{F}^{-1}(\tau)/(1 - \tau_c) \\ &\quad + \frac{\tau_c(1 - \tau)}{1 - \tau_c} \underset{t \sim T|C=c, t > c}{\mathbb{E}} [t] \\ &= -(1 - \tau) \underset{t \sim T|C=c, t \leq c}{\mathbb{E}} [t] + \frac{\tau_c(1 - \tau)}{1 - \tau_c} \underset{t \sim T|C=c, t > c}{\mathbb{E}} [t]. \end{aligned}$$

Since this value is the same for $S_{\text{Portnoy}}(\hat{F}, (z, \delta); w, \tau)$ and $S_{\text{Portnoy}}(F, (z, \delta); w, \tau)$, we have

$$\underset{t \sim T|C=c}{\mathbb{E}} [S_{\text{Portnoy}}(\hat{F}, (z, \delta); w, \tau)] = \underset{t \sim T|C=c}{\mathbb{E}} [S_{\text{Portnoy}}, (z, \delta); w, \tau)].$$

**Case 2.** We prove the case for $\max\{F^{-1}(\tau), \hat{F}^{-1}(\tau)\} < c$.

If $F^{-1}(\tau) \le \hat{F}^{-1}(\tau) < c$, then we have

$$
S_{\text{Portnoy}}(\hat{F}, (z, \delta); w, \tau)
$$

$$
= \begin{cases} \rho_\tau(\hat{F}^{-1}(\tau), t) & \text{if } t \le c, \\ w\rho_\tau(\hat{F}^{-1}(\tau), c) + (1-w)\rho_\tau(\hat{F}^{-1}(\tau), z_\infty) & \text{if } t > c, \end{cases}
$$

$$
= \begin{cases} (1-\tau)(\hat{F}^{-1}(\tau) - t) & \text{if } t \le \hat{F}^{-1}(\tau), \\ -\tau(\hat{F}^{-1}(\tau) - t) & \text{if } \hat{F}^{-1}(\tau) < t \le c, \\ -w\tau(\hat{F}^{-1}(\tau) - t) - (1-w)\tau(\hat{F}^{-1}(\tau) - t) & \text{if } t > c, \end{cases}
$$

$$
\ge \begin{cases} (1-\tau)(\hat{F}^{-1}(\tau) - t) & \text{if } t \le F^{-1}(\tau), \\ -\tau(\hat{F}^{-1}(\tau) - t) & \text{if } F^{-1}(\tau) < t \le \hat{F}^{-1}(\tau), \\ -\tau(\hat{F}^{-1}(\tau) - t) & \text{if } \hat{F}^{-1}(\tau) < t \le c, \\ -\tau(\hat{F}^{-1}(\tau) - t) & \text{if } t > c, \end{cases}
$$

$$
= \begin{cases} (1-\tau)(\hat{F}^{-1}(\tau) - t) & \text{if } t \le F^{-1}(\tau), \\ -\tau(\hat{F}^{-1}(\tau) - t) & \text{if } F^{-1}(\tau) < t, \end{cases}
$$

where we used $-w\tau(\hat{F}^{-1}(\tau) - t) \le w(1-\tau)(\hat{F}^{-1}(\tau) - t)$ when $F^{-1}(\tau) < t \le \hat{F}^{-1}(\tau)$ and $w \ge 0$ for the inequality.

If $\hat{F}^{-1}(\tau) \le F^{-1}(\tau) < c$, then we have

$$
S_{\text{Portnoy}}(\hat{F}, (z, \delta); w, \tau)
$$

$$
= \begin{cases} \rho_\tau(\hat{F}^{-1}(\tau), t) & \text{if } t \le c, \\ w\rho_\tau(\hat{F}^{-1}(\tau), c) + (1-w)\rho_\tau(\hat{F}^{-1}(\tau), z_\infty) & \text{if } t > c, \end{cases}
$$

$$
= \begin{cases} (1-\tau)(\hat{F}^{-1}(\tau) - t) & \text{if } t \le \hat{F}^{-1}(\tau), \\ -\tau(\hat{F}^{-1}(\tau) - t) & \text{if } \hat{F}^{-1}(\tau) < t \le c, \\ -w\tau(\hat{F}^{-1}(\tau) - t) - (1-w)\tau(\hat{F}^{-1}(\tau) - t) & \text{if } t > c, \end{cases}
$$

$$
> \begin{cases} (1-\tau)(\hat{F}^{-1}(\tau) - t) & \text{if } t \le \hat{F}^{-1}(\tau), \\ (1-\tau)(\hat{F}^{-1}(\tau) - t) & \text{if } \hat{F}^{-1}(\tau) < t \le F^{-1}(\tau), \\ -\tau(\hat{F}^{-1}(\tau) - t) & \text{if } F^{-1}(\tau) < t \le c, \\ -\tau(\hat{F}^{-1}(\tau) - t) & \text{if } t > c, \end{cases}
$$

$$
= \begin{cases} (1-\tau)(\hat{F}^{-1}(\tau) - t) & \text{if } t \le F^{-1}(\tau), \\ -\tau(\hat{F}^{-1}(\tau) - t) & \text{if } F^{-1}(\tau) < t, \end{cases}
$$

where we used $-w\tau(\hat{F}^{-1}(\tau) - t) > w(1-\tau)(\hat{F}^{-1}(\tau) - t)$ when $\hat{F}^{-1}(\tau) < t \le F^{-1}(\tau)$ and $w \ge 0$ for the inequality.

By Assumption 3.1, we have $\Pr(t \le F^{-1}(\tau)|C = c) = \Pr(t \le F^{-1}(\tau)) = \tau$ and $\Pr(F^{-1}(\tau) < t|C = c) = 1 - \tau$. Hence, we have

$$
\mathbb{E}_{t \sim T|C=c}[S_{\text{Portnoy}}(\hat{F}, (z, \delta); w, \tau)]
$$

$$
\ge \Pr(t \le F^{-1}(\tau)|C = c)(1-\tau)\hat{F}^{-1}(\tau) - (1-\tau)\mathbb{E}_{t \sim T|C=c, t \le F^{-1}(\tau)}[t]
$$

$$
- \Pr(F^{-1}(\tau) < t|C = c)\tau\hat{F}^{-1}(\tau) + \tau\mathbb{E}_{t \sim T|C=c, F^{-1}(\tau) < t}[t]
$$

$$
= -(1-\tau)\mathbb{E}_{t \sim T|C=c, t \le F^{-1}(\tau)}[t] + \tau\mathbb{E}_{t \sim T|C=c, F^{-1}(\tau) < t}[t].
$$

By using a similar argument, we have

$$
\mathbb{E}_{t \sim T|C=c}[S_{\text{Portnoy}}(F, (z, \delta); w, \tau)] = -(1-\tau)\mathbb{E}_{t \sim T|C=c, t \le F^{-1}(\tau)}[t] + \tau\mathbb{E}_{t \sim T|C=c, F^{-1}(\tau) < t}[t].
$$

Note that this equation holds with equality.

Hence, we have

$$\mathbb{E}_{t\sim T|C=c}[S_{\text{Portnoy}}(\hat{F},(z,\delta);w,\tau)] \geq \mathbb{E}_{t\sim T|C=c}[S_{\text{Portnoy}}(F,(z,\delta);w,\tau)].$$

**Case 3.** We prove the case for $F^{-1}(\tau) < c \leq \hat{F}^{-1}(\tau)$.

We have

$$
\begin{aligned}
S_{\text{Portnoy}}(\hat{F},(z,\delta);w,\tau) &= \begin{cases} \rho_\tau(\hat{F}^{-1}(\tau),t) & \text{if } t \leq c, \\ w\rho_\tau(\hat{F}^{-1}(\tau),c) + (1-w)\rho_\tau(\hat{F}^{-1}(\tau),z_\infty) & \text{if } t > c, \end{cases} \\
&= \begin{cases} (1-\tau)(\hat{F}^{-1}(\tau)-t) & \text{if } t \leq c, \\ w(1-\tau)(\hat{F}^{-1}(\tau)-c) - (1-w)\tau(\hat{F}^{-1}(\tau)-c) & \text{if } t > c, \end{cases} \\
&\geq \begin{cases} (1-\tau)(\hat{F}^{-1}(\tau)-t) & \text{if } t \leq F^{-1}(\tau), \\ -\tau(\hat{F}^{-1}(\tau)-t) & \text{if } F^{-1}(\tau) < t \leq c, \\ -\tau(\hat{F}^{-1}(\tau)-c) & \text{if } t > c, \end{cases}
\end{aligned}
$$

where we used $-w\tau(\hat{F}^{-1}(\tau)-t) \leq w(1-\tau)(\hat{F}^{-1}(\tau)-t)$ when $F^{-1}(\tau) < t \leq c \leq \hat{F}^{-1}(\tau)$ and $w \geq 0$, and $w(1-\tau)(\hat{F}^{-1}(\tau)-c) > -w\tau(\hat{F}^{-1}(\tau)-c)$ when $\hat{F}^{-1}(\tau) > t > c$ and $w \geq 0$ for the inequality. By Assumption 3.1, we have $\Pr(t \leq F^{-1}(\tau)|C=c) = \Pr(t \leq F^{-1}(\tau)) = \tau$, $\Pr(F^{-1}(\tau) < t \leq c|C=c) = \tau_c - \tau$, and $\Pr(t > c|C=c) = 1 - \tau_c$. Hence, we have

$$
\begin{aligned}
&\mathbb{E}_{t\sim T|C=c}[S_{\text{Portnoy}}(\hat{F},(z,\delta);w,\tau)] \\
&\geq \Pr(t \leq F^{-1}(\tau)|C=c)(1-\tau)\hat{F}^{-1}(\tau) - (1-\tau)\mathbb{E}_{t\sim T|C=c,t\leq F^{-1}(\tau)}[t] \\
&\quad -\Pr(F^{-1}(\tau) < t \leq c|C=c)\tau\hat{F}^{-1}(\tau) + \tau\mathbb{E}_{t\sim T|C=c,F^{-1}(\tau)<t}[t] \\
&\quad -\Pr(t > c|C=c)\tau\hat{F}^{-1}(\tau) + \tau c \\
&= -(1-\tau)\mathbb{E}_{t\sim T|C=c,t\leq F^{-1}(\tau)}[t] + \tau\mathbb{E}_{t\sim T|C=c,F^{-1}(\tau)<t\leq c}[t] + \tau c.
\end{aligned}
$$

By using a similar argument, we have

$$
S_{\text{Portnoy}}(F,(z,\delta);w,\tau) = \begin{cases} (1-\tau)(\hat{F}^{-1}(\tau)-t) & \text{if } t \leq F^{-1}(\tau), \\ -\tau(\hat{F}^{-1}(\tau)-t) & \text{if } F^{-1}(\tau) < t \leq c, \\ -\tau(\hat{F}^{-1}(\tau)-c) & \text{if } t > c, \end{cases}
$$

and so we have

$$\mathbb{E}_{t\sim T|C=c}[S_{\text{Portnoy}}(F,(z,\delta);w,\tau)] = -(1-\tau)\mathbb{E}_{t\sim T|C=c,t\leq F^{-1}(\tau)}[t] + \tau\mathbb{E}_{t\sim T|C=c,F^{-1}(\tau)<t\leq c}[t] + \tau c.$$

Note that this equation holds with equality.

Hence, we have

$$\mathbb{E}_{t\sim T|C=c}[S_{\text{Portnoy}}(\hat{F},(z,\delta);w,\tau)] \geq \mathbb{E}_{t\sim T|C=c}[S_{\text{Portnoy}}(F,(z,\delta);w,\tau)].$$

**Case 4.** We prove the case for $\hat{F}^{-1}(\tau) < c \leq F^{-1}(\tau)$.

We have

$$S_{\text{Portnoy}}(\hat{F}, (z, \delta); w, \tau)$$

$$= \begin{cases} \rho_\tau(\hat{F}^{-1}(\tau), t) & \text{if } t \leq c, \\ w\rho_\tau(\hat{F}^{-1}(\tau), c) + (1-w)\rho_\tau(\hat{F}^{-1}(\tau), z_\infty) & \text{if } t > c, \end{cases}$$

$$= \begin{cases} (1-\tau)(\hat{F}^{-1}(\tau) - t) & \text{if } t \leq \hat{F}^{-1}(\tau), \\ -\tau(\hat{F}^{-1}(\tau) - t) & \text{if } \hat{F}^{-1}(\tau) < t \leq c, \\ -w\tau(\hat{F}^{-1}(\tau) - c) - (1-w)\tau(\hat{F}^{-1}(\tau) - c) & \text{if } t > c, \end{cases}$$

$$> \begin{cases} (1-\tau)(\hat{F}^{-1}(\tau) - t) & \text{if } t \leq \hat{F}^{-1}(\tau), \\ (1-\tau)(\hat{F}^{-1}(\tau) - t) & \text{if } \hat{F}^{-1}(\tau) < t \leq c, \\ w(1-\tau)(\hat{F}^{-1}(\tau) - c) - (1-w)\tau(\hat{F}^{-1}(\tau) - c) & \text{if } t > c, \end{cases}$$

$$= \begin{cases} (1-\tau)(\hat{F}^{-1}(\tau) - t) & \text{if } t \leq c, \\ -\tau_c(1-\tau)(\hat{F}^{-1}(\tau) - c)/(1-\tau_c) & \text{if } t > c. \end{cases}$$

where we used $-w\tau(\hat{F}^{-1}(\tau) - t) > w(1-\tau)(\hat{F}^{-1}(\tau) - t)$ when $\hat{F}^{-1}(\tau) < t < c$ and $w \geq 0$, and $-w\tau(\hat{F}^{-1}(\tau) - c) > w(1-\tau)(\hat{F}^{-1}(\tau) - c)$ when $c > \hat{F}^{-1}(\tau)$ and $w \geq 0$ for the inequality, and $w = (\tau - \tau_c)/(1 - \tau_c)$ when $\tau_c \leq \tau$ for the last equality. By Assumption 3.1, we have $\Pr(t \leq c | C = c) = \Pr(t \leq c) = \tau_c$ and $\Pr(t > c | C = c) = 1 - \tau_c$. Hence, we have

$$\mathop{\mathbb{E}}_{t \sim T | C = c} [S_{\text{Portnoy}}(\hat{F}, (z, \delta); w, \tau)]$$

$$\geq \Pr(t \leq c | C = c)(1-\tau)\hat{F}^{-1}(\tau) - (1-\tau)\mathop{\mathbb{E}}_{t \sim T | C = c, t \leq c}[t]$$

$$- \Pr(t > c | C = c)\tau_c(1-\tau)\hat{F}^{-1}(\tau)/(1-\tau_c) + \frac{\tau_c(1-\tau)}{1-\tau_c}c$$

$$= -(1-\tau)\mathop{\mathbb{E}}_{t \sim T | C = c, t \leq c}[t] + \frac{\tau_c(1-\tau)}{1-\tau_c}c.$$

By using a similar argument, we have

$$\mathop{\mathbb{E}}_{t \sim T | C = c} [S_{\text{Portnoy}}(F, (z, \delta); w, \tau)] = -(1-\tau)\mathop{\mathbb{E}}_{t \sim T | C = c, t \leq c}[t] + \frac{\tau_c(1-\tau)}{1-\tau_c}c.$$

Note that this equation holds with equality.

Hence, we have

$$\mathop{\mathbb{E}}_{t \sim T | C = c} [S_{\text{Portnoy}}(\hat{F}, (z, \delta); w, \tau)] \geq \mathop{\mathbb{E}}_{t \sim T | C = c} [S_{\text{Portnoy}}(F, (z, \delta); w, \tau)].$$

$\square$

### A.2 Variant of Logarithmic Score

We show a proof of Theorem 4.6.

*Proof.* We consider a fixed $c \sim C$, and let $t$ be a sample obtained from $T$. Let $i$ be the index such that $\zeta_i \leq c < \zeta_{i+1}$. Since Assumption 3.1 holds, we have $\Pr(\zeta_j < t \leq \zeta_{j+1} | C = c) = \Pr(\zeta_j < t \leq \zeta_{j+1}) = F(\zeta_{j+1}) - F(\zeta_j) = f_j$ for any $j < i$, $\Pr(\zeta_i < t \leq c | C = c) = F(c) - F(\zeta_i)$, and

$\Pr(c < t | C = c) = \Pr(c < t) = 1 - F(c)$. Hence, we have

$$
\begin{aligned}
& \mathbb{E}_{t \sim T | C = c}[S_{\text{Cen-log}}(\hat{F}, (z, \delta); w, \{\zeta_i\}_{i=0}^B)] \\
& = \quad - \sum_{j < i} \Pr(\zeta_j < t \le \zeta_{j+1} | C = c) \log \hat{f}_j \\
& \quad - \Pr(\zeta_i < t \le c | C = c) \log \hat{f}_i \\
& \quad - \Pr(c < t | C = c) \left( w \log \hat{f}_i + (1 - w) \log(1 - \hat{F}(\zeta_{i+1})) \right) \\
& = \quad - \sum_{j < i} f_j \log \hat{f}_j \\
& \quad - (F(c) - F(\zeta_i)) \log \hat{f}_i \\
& \quad - (1 - F(c)) \left( w \log \hat{f}_i + (1 - w) \log(1 - \hat{F}(\zeta_{i+1})) \right) \\
& = \quad - \sum_{j \le i} f_j \log \hat{f}_j - (1 - F(\zeta_{i+1})) \log(1 - \hat{F}(\zeta_{i+1})),
\end{aligned}
$$

where we used $w = (F(\zeta_{i+1}) - F(c)) / (1 - F(c))$ for the last equality.

Hence, we have

$$
\begin{aligned}
& \mathbb{E}_{t \sim T | C = c}[S_{\text{Cen-log}}(\hat{F}, (z, \delta); w, \{\zeta_i\}_{i=0}^B)] - \mathbb{E}_{t \sim T | C = c}[S_{\text{Cen-log}}(F, (z, \delta); w, \{\zeta_i\}_{i=0}^B)] \\
& = \quad - \sum_{j \le i} f_j (\log \hat{f}_j - \log f_j) - (1 - F(\zeta_{i+1}))(\log(1 - \hat{F}(\zeta_{i+1})) - \log(1 - F(\zeta_{i+1}))) \\
& \ge \quad 0, \quad\quad\quad\quad\quad\quad\quad\quad\quad\quad\quad\quad\quad\quad\quad\quad\quad\quad\quad\quad\quad\quad\quad\quad (12)
\end{aligned}
$$

where we used the fact that the Kullback-Leibler divergence between two probability distributions is non-negative for the inequality. This means that the inequality

$$
- \sum_k p_k (\log \hat{p}_k - \log p_k) \ge 0
$$

holds for any two probability distributions $p_k$ and $\hat{p}_k$ and the equality holds only if $p_k = \hat{p}_k$ for all $k$. Here, we use an $(i + 2)$-dimensional vector $\boldsymbol{p} = (p_0, p_1, \ldots, p_{i+1})$, and we set $p_k = f_k$ for all $k \le i$ and we set $p_{i+1} = 1 - F(\zeta_{i+1})$. Note that the vectors $\boldsymbol{p}$ and $\hat{\boldsymbol{p}}$ constructed in this way is a probability distribution (i.e., $\sum_k p_k = 1$).

Since Inequality (12) holds for any $c \sim C$, we marginalize the inequality with respect to $C$, and we have

$$
\mathbb{E}_{t \sim T, c \sim C}[S_{\text{Cen-log}}(\hat{F}, (z, \delta); w, \{\zeta_i\}_{i=0}^B)] \ge \mathbb{E}_{t \sim T, c \sim C}[S_{\text{Cen-log}}(F, (z, \delta); w, \{\zeta_i\}_{i=0}^B)],
$$

which means that $S_{\text{Cen-log}}(\hat{F}, (z, \delta))$ is proper. Moreover, the equality holds only if $\hat{F} = F$, and therefore, $S_{\text{Cen-log}}(\hat{F}, (z, \delta))$ is *strictly* proper. $\qquad \square$

### A.3 VARIANT OF BRIER SCORE

We show a proof of Theorem 4.7.

*Proof.* We consider a fixed $c \sim C$, and let $t$ be a sample obtained from $T$. Let $i$ be the index such that $\zeta_i < c \le \zeta_{i+1}$. Assuming that Assumption 3.1 holds, we have $\Pr(\zeta_j < t \le \zeta_{j+1} | C = c) = \Pr(\zeta_j < t \le \zeta_{j+1}) = F(\zeta_{j+1}) - F(\zeta_j) = f_j$ for any $j < i$, $\Pr(\zeta_i < t \le c | C = c) = F(c) - F(\zeta_i)$,

and $\Pr(c < t | C = c) = \Pr(c < t) = 1 - F(c)$. Hence, we have

$$\mathbb{E}_{t \sim T | C = c}[S_{\text{Cen-Brier}}(\hat{F}, (z, \delta); \{w_i\}_{i=0}^{B-1}, \{\zeta_i\}_{i=0}^{B})]$$

$$= \sum_{j < i} \Pr(\zeta_j < t \le \zeta_{j+1} | C = c) \left( (1 - \hat{f}_j)^2 + \sum_{j \ne k} \hat{f}_k^2 \right)$$

$$+ \Pr(\zeta_i < t \le c | C = c) \left( (1 - \hat{f}_j)^2 + \sum_{j \ne k} \hat{f}_k^2 \right)$$

$$+ \Pr(c < t | C = c) \left( w_i (1 - \hat{f}_i)^2 + (1 - w_i) \hat{f}_i^2 + \sum_{j < i} \hat{f}_j^2 + \sum_{j > i} (w_j (1 - \hat{f}_j)^2 + (1 - w_j) \hat{f}_j^2) \right)$$

$$= \sum_{j < i} f_j \left( (1 - \hat{f}_j)^2 + \sum_{j \ne k} \hat{f}_k^2 \right) + (F(c) - F(\zeta_i)) \left( (1 - \hat{f}_j)^2 + \sum_{j \ne k} \hat{f}_k^2 \right)$$

$$+ (1 - F(c)) \left( w_i (1 - \hat{f}_i)^2 + (1 - w_i) \hat{f}_i^2 + \sum_{j < i} \hat{f}_j^2 + \sum_{j > i} (w_j (1 - \hat{f}_j)^2 + (1 - w_j) \hat{f}_j^2) \right)$$

$$= \sum_j (\hat{f}_j^2 - 2 f_j \hat{f}_j + 1),$$

where we used

$$w_i = \begin{cases} 0 & \text{if } \delta = 1 \text{ and } \zeta_{i+1} < z = t \\ 1 & \text{if } \delta = 1 \text{ and } \zeta_i < z = t \le \zeta_{i+1} \\ 0 & \text{if } z \le \zeta_i \end{cases}$$

for the first equality and

$$w_i = \begin{cases} (F(\zeta_{i+1}) - F(c))/(1 - F(c)) & \text{if } \delta = 0 \text{ and } \zeta_i < z = c \le \zeta_{i+1} \\ f_j/(1 - F(c)) & \text{if } \delta = 0 \text{ and } \zeta_{i+1} < z = c \end{cases}$$

for the last equality.

Hence we have

$$\mathbb{E}_{t \sim T | C = c}[S_{\text{Cen-Brier}}(\hat{F}, (z, \delta))] - \mathbb{E}_{t \sim T | C = c}[S_{\text{Cen-Brier}}(F, (z, \delta))]$$

$$= \sum_j (\hat{f}_j^2 - f_j^2 - 2 f_j (\hat{f}_j - f_j))$$

$$= \sum_j (\hat{f}_j - f_j)^2$$

$$\ge 0. \tag{13}$$

Note that the equality holds only if $\hat{f}_j = f_j$ holds for all $j$.

Since Inequality (13) holds for any $c \sim C$, we have

$$\mathbb{E}_{t \sim T, c \sim C}[S_{\text{Cen-Brier}}(\hat{F}, (z, \delta))] \ge \mathbb{E}_{t \sim T, c \sim C}[S_{\text{Cen-Brier}}(F, (z, \delta))],$$

which means that $S_{\text{Cen-Brier}}(\hat{F}, (z, \delta))$ is strictly proper. □

## B  ADDITIONAL EXPERIMENTS

In this section, we report the results of our additional experiments. The source codes used in our experiments are attached as the supplementary material.

In our experiments, we used the flchain, prostateSurvival, and support datasets, and we split the data points into training (60%), validation (20%), and test (20%). For each dataset, we divided the

Table 2: Comparison between two variants of the logarithmic score for $B = 8$

| Metric | Loss Function | flchain | prostateSurvival | support |
|---|---|---|---|---|
| $S_{\mathrm{Cen-log-simple}}$ | $S_{\mathrm{Cen-log}}$ | $6.4618 \pm 0.1204$ | $1.3460 \pm 0.0476$ | $1.5422 \pm 0.0704$ |
| | $S_{\mathrm{Cen-log-simple}}$ | $6.4176 \pm 0.1266$ | $1.3447 \pm 0.0451$ | $1.5368 \pm 0.0701$ |
| D-calibration | $S_{\mathrm{Cen-log}}$ | $\mathbf{0.0045} \pm 0.0004$ | $0.0002 \pm 0.0000$ | $0.0370 \pm 0.0032$ |
| | $S_{\mathrm{Cen-log-simple}}$ | $\mathbf{0.0127} \pm 0.0013$ | $0.0002 \pm 0.0001$ | $0.0349 \pm 0.0024$ |
| KM-calibration | $S_{\mathrm{Cen-log}}$ | $\mathbf{0.0048} \pm 0.0026$ | $0.0048 \pm 0.0028$ | $0.0057 \pm 0.0027$ |
| | $S_{\mathrm{Cen-log-simple}}$ | $\mathbf{0.0614} \pm 0.0081$ | $0.0083 \pm 0.0024$ | $0.0061 \pm 0.0033$ |

Table 3: Comparison between two variants of the logarithmic score for $B = 16$

| Metric | Loss Function | flchain | prostateSurvival | support |
|---|---|---|---|---|
| $S_{\mathrm{Cen-log-simple}}$ | $S_{\mathrm{Cen-log}}$ | $3.6774 \pm 0.0386$ | $1.2880 \pm 0.0247$ | $1.6017 \pm 0.0733$ |
| | $S_{\mathrm{Cen-log-simple}}$ | $3.6676 \pm 0.0424$ | $1.3447 \pm 0.0451$ | $1.6008 \pm 0.0731$ |
| D-calibration | $S_{\mathrm{Cen-log}}$ | $\mathbf{0.0005} \pm 0.0002$ | $0.0001 \pm 0.0000$ | $0.0147 \pm 0.0020$ |
| | $S_{\mathrm{Cen-log-simple}}$ | $\mathbf{0.0013} \pm 0.0004$ | $0.0002 \pm 0.0000$ | $0.0143 \pm 0.0021$ |
| KM-calibration | $S_{\mathrm{Cen-log}}$ | $0.0117 \pm 0.0046$ | $0.0142 \pm 0.0036$ | $0.0149 \pm 0.0080$ |
| | $S_{\mathrm{Cen-log-simple}}$ | $0.0162 \pm 0.0049$ | $0.0158 \pm 0.0063$ | $0.0158 \pm 0.0100$ |

Table 4: Comparison between two variants of the logarithmic score for $B = 32$

| Metric | Loss Function | flchain | prostateSurvival | support |
|---|---|---|---|---|
| $S_{\mathrm{Cen-log-simple}}$ | $S_{\mathrm{Cen-log}}$ | $1.5054 \pm 0.0508$ | $1.3608 \pm 0.0295$ | $1.8307 \pm 0.0452$ |
| | $S_{\mathrm{Cen-log-simple}}$ | $1.5059 \pm 0.0513$ | $1.3609 \pm 0.0301$ | $1.8296 \pm 0.0446$ |
| D-calibration | $S_{\mathrm{Cen-log}}$ | $0.0003 \pm 0.0001$ | $0.0001 \pm 0.0000$ | $0.0063 \pm 0.0009$ |
| | $S_{\mathrm{Cen-log-simple}}$ | $0.0003 \pm 0.0001$ | $0.0001 \pm 0.0000$ | $0.0062 \pm 0.0012$ |
| KM-calibration | $S_{\mathrm{Cen-log}}$ | $0.0206 \pm 0.0049$ | $0.0312 \pm 0.0084$ | $0.0299 \pm 0.0115$ |
| | $S_{\mathrm{Cen-log-simple}}$ | $0.0213 \pm 0.0049$ | $0.0343 \pm 0.0102$ | $0.0288 \pm 0.0127$ |

time interval $[0, z_{\max} + \epsilon)$, where $\epsilon = 10^{-3}$, into $B - 1$ equal-length intervals to get the thresholds $\{\zeta_i\}_{i=0}^{B}$ for distribution regression-based survival analysis, and we divided the unit interval $[0, 1]$ into $B - 1$ equal-length intervals to get the quantile levels $\{\tau_i\}_{i=0}^{B}$ for quantile regression-based survival analysis.

All our experiments were conducted on a virtual machine with an Intel Xeon CPU (3.30 GHz) processor without any GPU and 64 GB of memory running Red Hat Enterprise Linux Server 7.6. We used Python 3.7.4 and PyTorch 1.7.1 for the implementation.

We constructed a multi-layer perceptron (MLP) for our experiments. It consists of three hidden layers containing 128 neurons, and the number of outputs was $B$. The type of activation function after the hidden layer was the rectified linear unit (ReLU), and the activation function at the output node was softmax. We can satisfy the assumption that $\hat{F}(t)$ is a monotonically increasing continuous function by using it. In distribution regression-based survival analysis, each output of MLP estimates $\hat{f}_i = \hat{F}(\zeta_{i+1}) - \hat{F}(\zeta_i)$ for $i = 0, 1, \ldots, B - 1$. By using these outputs $\{\hat{f}_i\}_{i=0}^{B-1}$, we can calculate $\{\hat{F}(\zeta_i)\}_{i=0}^{B}$ and we can represent the function $\hat{F}(t)$ as a piecewise linear function connecting the values $\{\hat{F}(\zeta_i)\}_{i=0}^{B}$. Since $f_i > 0$ holds for all $i$, $\hat{F}(t)$ estimated in this way is a monotonically increasing continuous function. We can estimate $\hat{F}$ for quantile regression-based survival analysis by using a similar way.

First, we investigated the differences of the prediction performances between $S_{\mathrm{Cen-log}}$ (defined in Eq. (4)) and $S_{\mathrm{Cen-log-simple}}$ (defined in Eq. (5)) by using $S_{\mathrm{Cen-log-simple}}$, D-calibration, and KM-calibration as metrics. Tables 2–4 show the results for $B = 8, 16, 32$, respectively, where the bold numbers were used to emphasize the difference between two scoring rules. These results showed that the prediction performance of these two scoring rules were similar for the prostateSurvival and support datasets even for $B = 8$. However they showed different prediction performance for the flchain dataset for $B = 8$ and $B = 16$, but the performance difference were negligible for $B = 32$. Therefore, we used $B = 32$ in the other experiments in this paper.

Table 5: Prediction performances (lower is better) with various loss functions for $B = 32$

| Metric | Model | flchain | prostateSurvival | support |
|---|---|---|---|---|
| $S_{\text{Cen}-\log-\text{simple}}$ | DeepHit ($\alpha = 0.1$) | $1.5200 \pm 0.0398$ | $1.3644 \pm 0.0293$ | $1.8481 \pm 0.0453$ |
| | DeepHit ($\alpha = 1$) | $1.5858 \pm 0.0495$ | $1.3813 \pm 0.0318$ | $1.9996 \pm 0.0525$ |
| | DeepHit ($\alpha = 10$) | $2.0313 \pm 0.1648$ | $1.5688 \pm 0.0823$ | $2.3657 \pm 0.0441$ |
| | DRSA | $1.6783 \pm 0.0393$ | $1.4631 \pm 0.0273$ | $2.0342 \pm 0.0452$ |
| | S-CRPS | $2.0470 \pm 0.1575$ | $1.4589 \pm 0.0442$ | $2.1162 \pm 0.1095$ |
| | IPCW BS($t$) game | $1.9265 \pm 0.1093$ | $1.6413 \pm 0.0743$ | $2.3581 \pm 0.1604$ |
| | $S_{\text{Cen}-\log}$ | $1.5054 \pm 0.0508$ | $1.3608 \pm 0.0295$ | $1.8307 \pm 0.0452$ |
| | $S_{\text{Cen}-\text{Brier}}$ | $1.5137 \pm 0.0557$ | $1.3680 \pm 0.0291$ | $1.8467 \pm 0.0448$ |
| | $S_{\text{Cen}-\text{RPS}}$ | $1.6737 \pm 0.0821$ | $1.4821 \pm 0.0639$ | $2.1036 \pm 0.1012$ |
| | $S_{\text{Portnoy}}$ | $1.6641 \pm 0.0518$ | $1.4352 \pm 0.0420$ | $2.0645 \pm 0.0455$ |
| D-calibration | DeepHit ($\alpha = 0.1$) | $0.0005 \pm 0.0002$ | $0.0001 \pm 0.0000$ | $0.0056 \pm 0.0009$ |
| | DeepHit ($\alpha = 1$) | $0.0008 \pm 0.0003$ | $0.0003 \pm 0.0001$ | $0.0062 \pm 0.0010$ |
| | DeepHit ($\alpha = 10$) | $0.0138 \pm 0.0046$ | $0.0064 \pm 0.0035$ | $0.0179 \pm 0.0053$ |
| | DRSA | $0.0043 \pm 0.0011$ | $0.0047 \pm 0.0004$ | $0.0057 \pm 0.0006$ |
| | S-CRPS | $0.0032 \pm 0.0005$ | $0.0018 \pm 0.0004$ | $0.0072 \pm 0.0011$ |
| | IPCW BS($t$) game | $0.0022 \pm 0.0006$ | $0.0083 \pm 0.0018$ | $0.0060 \pm 0.0008$ |
| | $S_{\text{Cen}-\log}$ | $0.0003 \pm 0.0001$ | $0.0001 \pm 0.0000$ | $0.0063 \pm 0.0009$ |
| | $S_{\text{Cen}-\text{Brier}}$ | $0.0004 \pm 0.0002$ | $0.0001 \pm 0.0000$ | $0.0071 \pm 0.0009$ |
| | $S_{\text{Cen}-\text{RPS}}$ | $0.0005 \pm 0.0003$ | $0.0010 \pm 0.0005$ | $0.0045 \pm 0.0011$ |
| | $S_{\text{Portnoy}}$ | $0.0071 \pm 0.0031$ | $0.0055 \pm 0.0041$ | $0.0237 \pm 0.0037$ |
| KM-calibration | DeepHit ($\alpha = 0.1$) | $0.0264 \pm 0.0071$ | $0.0418 \pm 0.0139$ | $0.0249 \pm 0.0067$ |
| | DeepHit ($\alpha = 1$) | $0.0362 \pm 0.0084$ | $0.0599 \pm 0.0341$ | $0.0545 \pm 0.0110$ |
| | DeepHit ($\alpha = 10$) | $0.2077 \pm 0.0543$ | $0.4937 \pm 0.1772$ | $0.4273 \pm 0.1188$ |
| | DRSA | $0.1929 \pm 0.0135$ | $0.1845 \pm 0.0050$ | $0.2103 \pm 0.0162$ |
| | S-CRPS | $0.2759 \pm 0.1279$ | $0.6414 \pm 0.3043$ | $0.4090 \pm 0.1499$ |
| | IPCW BS($t$) game | $0.2770 \pm 0.0789$ | $0.4246 \pm 0.0841$ | $0.5325 \pm 0.1342$ |
| | $S_{\text{Cen}-\log}$ | $0.0206 \pm 0.0049$ | $0.0312 \pm 0.0084$ | $0.0299 \pm 0.0115$ |
| | $S_{\text{Cen}-\text{Brier}}$ | $0.0268 \pm 0.0071$ | $0.0324 \pm 0.0090$ | $0.0492 \pm 0.0125$ |
| | $S_{\text{Cen}-\text{RPS}}$ | $0.1553 \pm 0.0349$ | $0.5931 \pm 0.3846$ | $0.2668 \pm 0.1192$ |
| | $S_{\text{Portnoy}}$ | $0.0434 \pm 0.0067$ | $0.1895 \pm 0.1413$ | $0.0809 \pm 0.0381$ |

Next, we computed the prediction performance of several loss functions used in the state-of-the-art neural network models. The loss function of DeepHit (Lee et al., 2018) consists of two terms. The first term is equal to the extension of the logarithmic score $S_{\text{Cen}-\log-\text{simple}}$, and the second term is used to improve a ranking metric between patients. The parameter $\alpha$ is used to control the balance between these two terms, and the weight for the second term is increased by using a large $\alpha$. The loss function of DRSA (Ren et al., 2019) can also be seen as a variant of logarithmic score, and we set $\alpha = 0.25$ for the parameter. S-CRPS (Avati et al., 2019) is a variant of the ranked probability score, but Rindt et al. (2022) showed that this scoring rule is not proper in terms of theory of scoring rules. We also implemented the IPCW BS($t$) game model, which is proposed in (Han et al., 2021). Table 5 shows the results. The prediction performance of DeepHit degraded by using a large $\alpha$, which means that it is better to use $S_{\text{Cen}-\log-\text{simple}}$ by setting $\alpha = 0$. The other prediction models did not outperform $S_{\text{Cen}-\log}$ and $S_{\text{Cen}-\text{Brier}}$.

Finally, we show an ablation study on the training models with and without the EM algorithm. Figure 3 shows the average survival functions for $B = 32$, which means that the average of $\overline{F}(t) = 1 - F(t)$ for all patients in test dataset were shown. The parameter $w$ (or $\{w_i\}_{i=0}^{B-1}$) is included in the computation of the gradient in the neural network training of the prediction model *without* the EM algorithm, whereas the prediction model with the EM algorithm handles the parameter as a constant. The actual survival functions were estimated by the Kaplan-Meier estimator. These results showed that the average predictions for the extension of the logarithmic score were close to the Kaplan-Meier curve regardless of the use of the EM algorithm for the three datasets. As for the other three estimators, the average predictions with the EM algorithm were closer than those without the EM algorithm to the Kaplan-Meier curves. These results mean that we need the EM algorithm except for the extension of the logarithmic score.

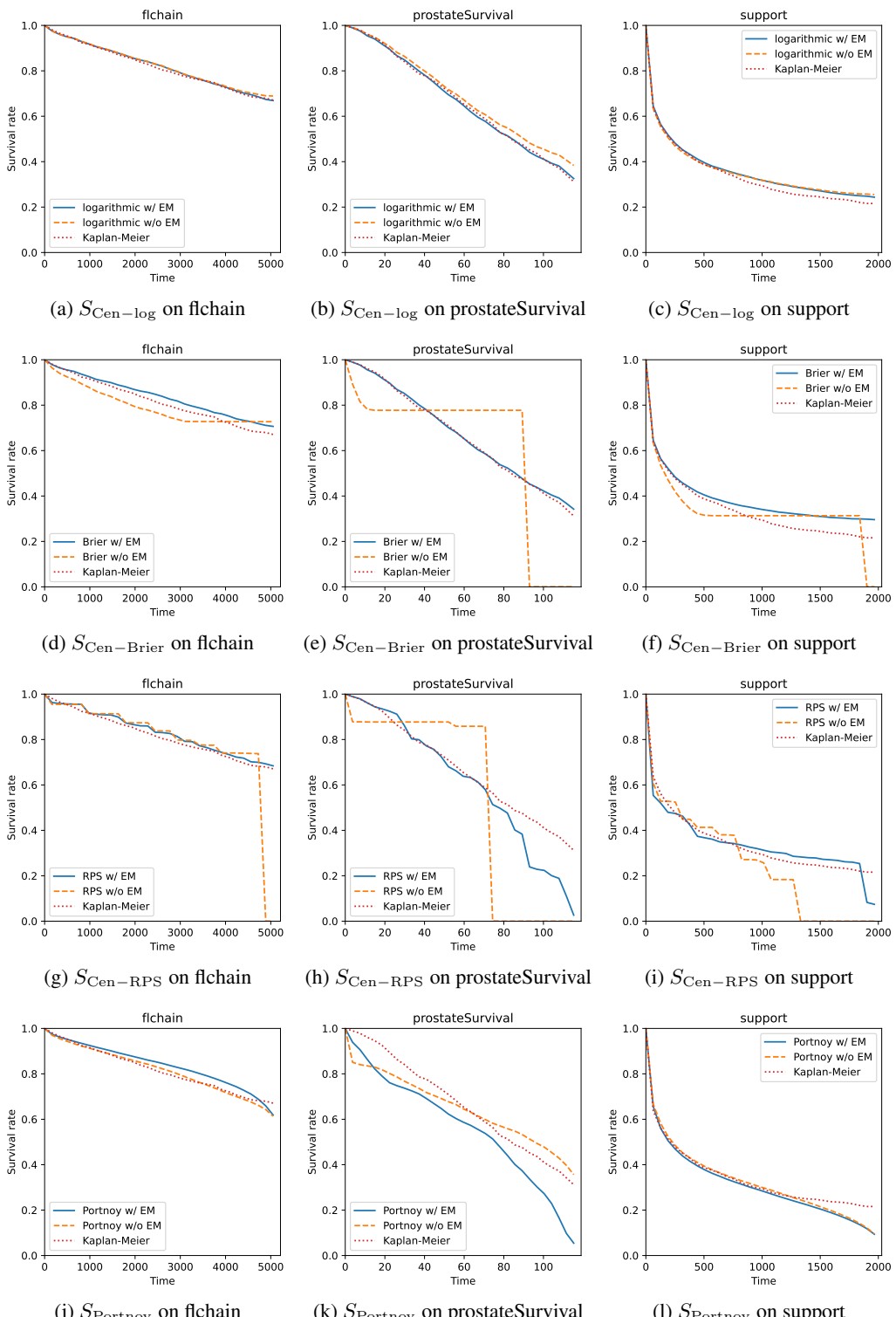

Figure 3: Comparisons of average survival functions with and without EM algorithm for $B = 32$

