# OpenReview forum: "Proper Scoring Rules for Survival Analysis"
_ICLR.cc/2023/Conference — Submitted to ICLR 2023_

### Official Review · Reviewer_hCyF · 2022-10-23

**Confidence:** 4
**Correctness:** 3
**Technical Novelty And Significance:** 3
**Empirical Novelty And Significance:** 2
**Recommendation:** 6

**Clarity, Quality, Novelty And Reproducibility:**

I can understand the paper easily.  The contribution is novel as far as I know.

**Strength And Weaknesses:**

Strength:
1. Strictly proper scoring rules are important for both training and evaluation in survival analysis. Due to the existence of censoring, strictly proper scoring rules are hard to find. The common training objectives are usually not strictly proper. The authors tackle this difficulty and propose four new extensions of existing objectives.

2. The authors also propose another evaluation metric for calibration based on the KL-divergence between Kaplan-Meier estimator and the proposed estimation.

Weakness:
1. The authors propose four strictly proper scoring rules under the condition parameters w are correct. However, unless we know the true distribution. But the true distribution is never known. So the authors need to discuss how hard we can get w in practice. And if we cannot get the w correctly, is using wrongly estimated w based "strictly proper scoring rule" still better than using those non-strictly proper ones?

2. In experiments, the authors only compare four new objectives. The authors have not compare with training with the non-proper version of the proposed algorithm. To make the things easier to compare with, the authors can use synthetic distributions, in which they can get the true distribution/true parameters w in the objectives, and the authors test the performance using the strictly proper scoring without estimation loss from w. The authors should also include some common training objectives, for example, likelihood during the comparison.

3. The authors should consider some other common evaluation metrics, for example, concordance. Though concordance is not a proper scoring rule, but it has a practical meaning in survival analysis, i.e., the patient with high risk should be estimated with high risk. This is helpful when allocating medical resources.

4. Closely related to the first point. Are the proposed objectives really helpful in training deep models? Deep models may not find the global optima in optimization. Finding good local optima is more important. So are "strictly proper scoring rules" really an important point? Will non-proper rule be better for optimization. The authors do not need to answer this in details. This is a hard topic.


Minor:
1. In introduction, bullet point Brier score, "show that neither of them are not proper in terms of the theory of scoring rules." I think the authors may want only one negation here. maybe remove "not".
2. I think when the authors define four new objectives, they probably need to define with the expectation. For example, the expectation of the Portnoy’s estimator is strictly proper but not the estimator itself.

**Summary Of The Paper:**

The authors propose four extensions of training objectives in survival analysis. Under some conditions, these metrics are strictly proper scoring rules. The main contributions are four new strictly proper objectives.

**Summary Of The Review:**

I think the authors deal with an interesting and important topic: proper scoring rules in survival analysis. They also made some contribution to propose new ones. But I think more evaluations need to be done as mentioned in weakness. I am happy to raise my score if the authors can address my point both theoretically and empirically in the weakness part.

---

> ### Author Response · Authors · 2022-11-19
> **Comments for your review**
>
> Thank you for your comments.  We have noticed that some important points in our paper could not be conveyed appropriately, and so we would like to explain them in this comment.
>
>
> > The authors propose four strictly proper scoring rules under the condition parameters w are correct. However, unless we know the true distribution. But the true distribution is never known. So the authors need to discuss how hard we can get w in practice. And if we cannot get the w correctly, is using wrongly estimated w based "strictly proper scoring rule" still better than using those non-strictly proper ones?
>
> When we obtain an estimation which is seemingly different from the true probability distribution, we can itemize possible reasons for the wrong estimation.  For example, if we use a scoring rule, the possible reasons are:
> 1. We used an inappropriate loss function (i.e., a scoring rule).
> 2. We failed to find a correct parameter $w$.
> 3. The estimation model (e.g., neural network) could not find the (global) optimum solution.
> 4. Other reasons (e.g., overfitting of the estimation model).
>
> One of our contributions is that we show that we can remove Possibility 1 if we use one of the proper scoring rules we proved in our paper.  This fact enables us to focus on investigating the other possibilities 2-4.
>
> Moreover, if we use $S_{\rm Cen-simple-log}$, which is one of the proper scoring rules, we can also remove Possibility 2, because it does not include the parameter $w$.  It is similar to the extension of the logarithmic score $S_{\rm Cen-cont-log}$ proved in [Rindt et al., 2022], but one of our contributions is that we clarified the implicit assumption in the proof of [Rindt et al., 2022] that $B$ is large enough.  We also show how to find an appropriate $B$ in the experiments in Appendix B (see Tables 2-4).
>
>
>
> > In experiments, the authors only compare four new objectives. The authors have not compare with training with the non-proper version of the proposed algorithm.
>
> In case you have missed the Appendix, we would like to let you know that Table 5 shows our comparisons against the state-of-the-art methods, which use non-proper loss functions.
>
>
>
>
>
> > The authors should consider some other common evaluation metrics, for example, concordance. Though concordance is not a proper scoring rule, but it has a practical meaning in survival analysis, i.e., the patient with high risk should be estimated with high risk. This is helpful when allocating medical resources.
>
> To answer this comment, we should be aware the distinction between two different formulations of survival analysis.
> 1. One of them is to formulate the survival analysis as estimating a hazard rate (i.e., a scalar value to represent a risk) for each patient.  This formulation is based on the proportional hazard assumption, and it is widely used in survival analysis (including the classical Cox model [Cox, 1972]).  We agree that estimating high risk patients has important practical meaning.  For such applications, we think that we should use the proportional hazard assumption, because the definition of the "risk" of a patient is clear.  When we use this formulation, there is no problem on using the concordance index.
> 2. Another formulation of survival analysis is to estimate the CDF $F$ of each patient.  In this paper, we consider this formulation.  A drawback of this formulation is that the definition of the "risk" of a patient is unclear because the estimation is a CDF $F$ and it is not a scalar value.  Therefore, the meaning of "concordance" is unclear in this formulation, and there is no unanimous agreement on the definition on the extension of the concordance index.  Indeed there are some variants of the concordance index for this formulation of survival analysis, and the problems on using these variants are extensively discussed in [Sonabend et al., 2022].
>
> We would also like to note that Formulation 2 (i.e., w/ censored data points) is a natural extension of uncertainty quantification (i.e., w/o censored data points).  In uncertainty quantification, we usually use scoring rules that are proved to be proper and calibration metrics as evaluation metrics.  This is why we use the proper scoring rule and the calibration metrics in our experiments.

---

### Official Review · Reviewer_rsGQ · 2022-10-25

**Confidence:** 3
**Correctness:** 4
**Technical Novelty And Significance:** 3
**Empirical Novelty And Significance:** 3
**Recommendation:** 6

**Clarity, Quality, Novelty And Reproducibility:**

[Clarity]

The language part is generally clear. But some math part is not clear:

1. Introduction, "Brier score" item: "show that neither of them are not proper in terms of the theory of scoring rules" --> "neither of them is proper"

2. Related Work: "Since we do not require the theory of scoring rules under this assumption" --> "Since the theory of scoring rules does not need this assumption".

3. Assumption 3.1: What is feature X? Does X affect the distribution of T? Namely, is the CDF of T still equal to F(t) after conditioning on X? (I think it is, according to your proof of Theorem 4.5). Also, does X affect the distribution of C? If X does not affect the distributions of T and C then why do you define X?

4. Page 4: "Therefore, the inequality $S(\hat F_1, y) < S(\hat F_2, y)$ means that ...". Should be the expectation of S.

5. Equation (4): should "$w$" be "$w_i$" because it depends on i. Also, what if $\zeta_{i+1}$ is very large, so that $F(\zeta_{i+1})$ is close to $F(c)$ and $w_i$ is not close to 0?

6. Equation (6): $f_i$ -> $\hat f_i$

7. Equation (9): should "1" be "$\hat F(\zeta)$"?


[Novelty]

As written in [Weakness], I cannot access the novelty of this paper due to lack of comparison with previous works.

**Strength And Weaknesses:**

[Strength]

Survival analysis is definitely an important problem to study, and the use of proper scoring rules to solve this problem is also well motivated. The authors prove that the proper scoring rules adapted to the survival problem are proper under some condition. This is an interesting result.

[Weakness]

However, I can't really tell the novelty of this work. Just as the authors mention, many works like (Portnoy, 2003), (Pearce et al, 2022) already proposed to use proper scoring rules to solve the survival analysis problem. Although the authors claim to be the first to prove that Portnoy's estimator (which is adapted from Pinball loss) is proper, this conclusion is under the strong assumption that the parameter w (which depends on the unknown distribution F(t)) is known. Under this assumption, the conclusion that Portnoy's estimator (and other three adapted scores) is proper is not surprising. It seems to me that the most challenging part of using scoring rules to solve survival analysis is how to estimate the parameter w, instead of proving properness. The methods to estimate w in this paper, however, basically follow from previous works.

The relationship with previous works is unclear. Introduction says that Rindt et al (2022) show that variants of Brier score and ranked probability score are not proper, but the variants in this paper are claimed to be proper (if w is known). Are they different variants? Also, did Portnoy (2003) or Pearce et al (2022) already prove that Portnoy's estimator is proper if we know w?


[Feedback, not weakness]

It might help to include some results in Appendix B (additional experiments) in the body, in particular, the comparison between the proper scoring rules in this paper and the custom loss functions in other state-of-the-art neural networks. One motivation for the authors to study properness of scoring rules in the survival analysis problem is because the scoring rules are used as loss functions in training neural networks, where properness is desired. So, the empirical comparison between proper scoring rules and other custom loss functions is worth including.

**Summary Of The Paper:**

The paper studies how to use proper scoring rules to solve the survival analysis problem, where we want to estimate the distribution F(t) of a random variable T using possibly censored data, which are samples of Z = min{T, C} and whether Z = T or C. While any proper scoring rule can be used as a loss function to train a model (e.g., a neural network) to estimate a distribution using uncensored data, to train on censored data the original proper scoring rule needs to be adapted. The authors adapt four proper scoring rules (Pinball, Logarithmic, Brier, and Ranked Probability Score), and theoretically prove that these adapted socring rules are proper, under the assumption that a parameter w (that depends on the distribution F(t)) is known.

In practice, however, the parameter w is unknown, so the authors use EM and CQRNN algorithms to choose w and train the model together. Experiments show that the adapted logarithmic scoring rule is the best.

**Summary Of The Review:**

I currently recommend weak reject due to the lack of novelty and clarity. But if the authors can provide more contexts, like comparisons with previous works, to demonstrate the novelty of their works, I might change my opinion.

---

> ### Author Response · Authors · 2022-11-19
> **Comments for your review**
>
> First of all, thank you for your comments, especially for the comments in the [Clarity] section.
>
> > However, I can't really tell the novelty of this work.... It seems to me that the most challenging part of using scoring rules to solve survival analysis is how to estimate the parameter w, instead of proving properness.
>
> We think that you have already understood our contributions (unconsciously?).  Primary message of this paper is to tell that "the most challenging part of using scoring rules to solve survival analysis is how to estimate the parameter w".
>
> Since you might underestimate the importance of giving a proof of properness, we would like to emphasize that the most important thing in uncertainty quantification is to give a proof of properness.  Whereas it is easy to design a loss function $l(\hat{y}, y)$ for an estimation $\hat{y}$ and a data point $y$ in the standard regression, it is not easy to design an appropriate loss function (scoring rule) $S(\hat{F}, y)$ between an estimation $\hat{F}$ of CDF and a single data point $y$ in uncertainty quantification. This is why the theory of scoring rules have been established for uncertainty quantification and why we usually use a scoring rule with a proof of properness in uncertainty quantification.
>
> While the variants of Brier score [Graf et al., 1999] for survival analysis also depend on unknown parameter $G$ (similar to the parameter $w$ in our extension of Brier score), it is not proven that minimizing the variants of Brier score would yield true probability distribution even if we knew true $G$.  In contrast, we prove that our extension of Brier score is proper under the condition that $w$ is correct, and minimizing $S_{\rm Cen-Brier}$ would yield true probability distribution if we knew true $w$.
>
> We would also like to remind you that Table 1 shows that the four proper scoring rules have different prediction performance.  These differences indicate that the difficulty of estimating correct $w$ depends on the scoring rule.  Therefore, our recommendation is to use $S_{\rm Cen-log}$ or $S_{\rm Cen-Brier}$ as a scoring rule rather than seeking for a new algorithm to estimate $w$, although it is still an interesting research direction to find a new algorithm to estimate correct $w$ for $S_{\rm Portnoy}$ and $S_{\rm Cen-RPS}$.
>
>
>
> > The relationship with previous works is unclear. Introduction says that Rindt et al (2022) show that variants of Brier score and ranked probability score are not proper, but the variants in this paper are claimed to be proper (if w is known). Are they different variants?
>
> They are different.  We would like to clarify that our contributions are (i) a novel extension of the Brier score for survival analsis with the proof of properness, and (ii) the experimental results showing that our extension of Brier score is comparable to the proper variant of the logarithmic score [Rindt et al., 2022] and is superior to IPCW BS(t) game [Han et al., 2021], which uses the other (non-proper) variant of Brier score [Graf et al., 1999] (see Table 5).
>
>
> > Also, did Portnoy (2003) or Pearce et al (2022) already prove that Portnoy's estimator is proper if we know w?
>
> No.

---

> > ### Comment · Reviewer_rsGQ · 2022-12-02
> > **Raise score to 6**
> >
> > The authors' response resolves my concern about the novelty of this work, so I raised the score to 6.

---

> > > ### Author Response · Authors · 2022-12-05
> > > **Thank you for raising the score!**
> > >
> > > Thank you for raising the score!

---

### Official Review · Reviewer_PxCN · 2022-10-31

**Confidence:** 3
**Correctness:** 3
**Technical Novelty And Significance:** 3
**Empirical Novelty And Significance:** 2
**Recommendation:** 5

**Clarity, Quality, Novelty And Reproducibility:**

The paper is quite clear. However, given the parameter set w is so central in this paper I would have liked it explained better. The first encounter with the parameter w is after equation 2 and I find the description quite terse.

**Strength And Weaknesses:**

It is not obvious from the paper how explanatory covariates can be integrated. The parameter vector which is central in the correctness of the deduced proofs is distribution dependent. For each covariate combination, the values for w changes. Therefore, I cannot think of applying what this paper proposes in a survival analysis application in which the sample is not thought to be drawn from a homogeneous population.

The experiments are not shedding any light on what I mentioned above. Table 1 has several metrics for which several loss functions have been tried but the author is left guessing what to take out of it. Since this paper is concerned with survival analysis, it would have been better to show that the metrics that this paper proposes to be proper, when minimized, prove to be better than those normally minimized for which such guarantees do not exist.

**Summary Of The Paper:**

This work is after finding proper scoring rules in survival analysis. They have a parameter vector w whose true specification underpins the proofs their provide for the discussed scoring rules being proper. They approximate the parameter vector w using an EM algorithm which can in turn be plugged into their scoring rule.

**Summary Of The Review:**

I have doubts about the applicability of this work. It is not obvious to me how this work can be applied to a real-world scenario where explanatory covariates exist and the population from which the data is sampled is not homogeneous. The experiments did not seem to counter my current understanding.

---

> ### Author Response · Authors · 2022-11-19
> **Comments on your review**
>
> > It is not obvious from the paper how explanatory covariates can be integrated.
>
> Thank you for your comments.  We should have explained on your concern in our paper.  We use a standard approach of applying a scoring rule to uncertainty quantification (i.e., quantile regression and distribution regression), and we do not assume that there is a homogeneous population.
>
> On the contrary, we assume that there is (an unknown) probability distribution for each covariate.  More specifically, given a data point $(x,y)$, we formulate uncertainty quantification as estimating the true CDF $F_{x}$ for each $x$, and we compute the value of scoring rule $S(G_{x},y)$ for an estimation $G_{x}$. (We use $G_{x}$ because it seems to me that we cannot add a hat on $F_{x}$ here.) Since we are given a set of data points $D=\{ (x_{i},y_{i}) \}$, we formulate uncertainty quantification as minimizing $\sum_{i=1}^{n} S(G_{x_{i}}, y_{i})$.
>
> We have noticed that the first three lines of page 5 (more specifically, the sentence starting from "Therefore, quantile regression can be formulated as....") is misleading.  We are thinking about rephrasing or removing this sentence.
>
> > Since this paper is concerned with survival analysis, it would have been better to show that the metrics that this paper proposes to be proper, when minimized, prove to be better than those normally minimized for which such guarantees do not exist.
>
> In case you have missed the Appendix, we would like to let you know that Table 5 shows our comparisons using the metrics proposed in the state-of-the-art papers (i.e., the variant of the logarithmic score [Rindt et al., 2022] and D-calibration [Haider et al., 2020]).

---

### Decision · Program_Chairs · 2023-01-20

**Decision:**

Reject

**Justification For Why Not Higher Score:**

The paper is borderline.  The work is very specialised and its not clear at all it belongs in ICLR.  Its a very nice contribution to statistical theory.

**Justification For Why Not Lower Score:**

N/A

**Metareview: Summary, Strengths And Weaknesses:**

The paper is a theoretical study of scoring rules for survival analysis, including some empirical work.  The consider standard scoring rules such as logarithmic and Brier, and adaptations for survival analysis.  The reviewers pointed out a number of issues and the authors addressed quite a few of these.

The contribution and strength of the paper is to present extensions to the standard scoring rules and does some evaluations.  The paper gives a very nice contribution to statistical theory.

The review raised a number of issues with the paper and the authors clarified some of these but its not clear to what degree this prompted appropriate rewrites of the paper.   This is specialised work and would be excellent for a conference like AI & Stats.

**Summary Of Ac-Reviewer Meeting:**

N/A